# The long-standing relationship between Paramagnetic NMR and Iron-Sulfur proteins: the mitoNEET example. An old method for new stories or the other way around?

Francesca Camponeschi[1‡], Angelo Gallo[2‡], Mario Piccioli[1,3*], and Lucia Banci[1,3*]

[1] Consorzio Interuniversitario Risonanze Magnetiche MetalloProteine, Sesto Fiorentino, I-50019, Italy

[2] Department of Pharmacy, University of Patras, Patras, GR-26504, Greece

[3] Magnetic Resonance Center and Department of Chemistry, University of Florence, Sesto Fiorentino, I-50019, Italy

[‡]These authors contributed equally

*Correspondence to*: Lucia Banci (banci@cerm.unifi.it); Mario Piccioli (piccioli@cerm.unifi.it)

**Abstract.**

Paramagnetic NMR spectroscopy and iron-sulfur (Fe–S) proteins have maintained a synergic relationship for decades. Indeed, the hyperfine shifts with their temperature dependencies and the relaxation rates of nuclei of cluster-bound residues have been extensively used as a fingerprint of the type and of the oxidation state of the Fe–S cluster within the protein frame. The identification of NMR signals from residues surrounding the metal cofactor is crucial for understanding the structure-function relationship in Fe–S proteins, but it is generally impaired in standard NMR experiments by paramagnetic relaxation enhancement due to the presence of the paramagnetic cluster(s). On the other hand, the availability of systems of different size and stability has, over the years, stimulated NMR spectroscopists to exploit iron-sulfur proteins as paradigmatic cases to develop experiments, models and protocols. Here, the cluster binding properties of human mitoNEET have been investigated by one-dimensional and two-dimensional $^1$H diamagnetic and paramagnetic NMR, in its oxidized and reduced states. The NMR spectra of both oxidation states of mitoNEET appeared to be significantly different from those reported for previously investigated $[Fe_2S_2]^{2+/+}$ proteins. The protocol we have developed in this work conjugates spectroscopic information arising from "classical" paramagnetic NMR with an extended mapping of the signals of residues around the cluster which can be taken, even before the sequence specific assignment is accomplished, as a finger print of the protein region constituting the functional site of the protein. We show how the combined use of 1D NOE experiments, $^{13}$C direct-detected experiments, and double and triple resonance experiments tailored using $R_1$ and/or $R_2$-based filters, significantly reduces the "blind" sphere of the protein around the paramagnetic cluster. This approach provided a detailed description of the unique electronic properties of mitoNEET, which are responsible for its biological function. Indeed, the NMR properties suggested that the specific electronic structure of the cluster possibly drives the functional properties of different $[Fe_2S_2]$ proteins.

# 1 Introduction

After 40 years of a life-long relationship, iron-sulfur (Fe–S) proteins and paramagnetic NMR still maintain an active and fruitful "*liaison*". What makes them still connected one-another, which secrets are yet to be revealed? And, last but not least, which is, between the two, the one that better counteracts the effect of time passing, maintaining itself charming and interesting?

It is a story with many players and scenarios. Indeed, the first NMR spectra of Fe–S proteins date back to 1970, when W. D. Phillips, M. Poe and C. C. McDonald, published, in a few months period, the NMR spectra of: i) the two $[Fe_4S_4]^{2+}$ clusters ferredoxin from *C. Pasteurianum* (Poe et al., 1970), ii) the single $Fe^{3+}$ ion rubredoxin (Phillips et al., 1970a), iii) *C. vinosum* $[Fe_4S_4]$ HiPIP in both oxidation states (Phillips et al., 1970b), and, iv) parsley and spinach $[Fe_2S_2]$ cluster ferredoxins again in both oxidation states (Poe et al., 1971). Combined with Mössbauer, EPR and magnetic susceptibility data (Dunham et al., 1971), the chemical shift properties of the paramagnetically shifted signals and their temperature dependencies were used to propose, with alternate fortune, models for the type of the Fe–S clusters and of their electronic structure within these proteins. This series of papers is a landmark for both NMR of paramagnetic systems and for Fe–S proteins. Only one year earlier, the first interpretation of the NMR spectra of paramagnetic proteins appeared for cytochrome c (Kowalsky, 1965; McDonald et al., 1969; Wüthrich, 1969), and a very few articles were available on paramagnetic NMR spectra of transition metal complexes (Holm et al., 1966; La Mar and Sacconi, 1968; Sacconi and Bertini, 1966). The first NMR spectra of non-heme metalloproteins showed everyone the huge potential of NMR spectroscopy, capable to combine, on the one hand the information on the electronic structure of the paramagnetic center and on the other hand, its unique ability to identify individual hydrogen atoms within the protein frame. These features were extremely attractive for biochemists and biophysicists engaged into the understanding of Fe–S proteins. It was therefore soon clear that NMR spectroscopy could provide very useful contributions to the description of these systems (Beinert et al., 1997; Beinert and Albracht, 1982). The playground opened!

For about 25 years, one dimensional NMR experiments provided a sensitive fingerprint to address the type of Fe–S cluster present in a protein and its oxidation state. Eventually, the combination of NMR, EPR, Mössbauer and optical spectroscopies succeeded to convert a contest for the "best fingerprint technique" into a synergy of complementary spectroscopic tools (Garcia-Serres et al., 2018; Hagen, 2018). Electronic structure, electron transfer properties, magnetic couplings among the cluster iron ions, role of hydrogen bonds surrounding the cluster, driving factors of valence localization/delocalization, have been some of the major aspects successfully described (Banci et al., 1993; Bertini et al., 1997; Gaillard et al., 2002; Johnson et al., 2005; Nettesheim et al., 1983; Oh and Markley, 1990).

A tango-relationship has been maintained for decades: on the one hand the NMR information provided a deeper understanding on the structure, the reactivity, the stability, and the interaction patterns of Fe–S proteins; on the other hand, the availability of

metalloproteins of different size and stability stimulated biomolecular NMR spectroscopists to develop experiments, models and protocols. Many advancements in NMR contributed to this synergy: 1D NOE experiments (Dugad et al., 1990), the first solution structure of a paramagnetic metalloprotein (Banci et al., 1994), the use of $^{13}$C and $^{15}$N direct-detected experiments (Kostic et al., 2002; Machonkin et al., 2002), the synergy of several research groups involved into structural proteomics (Ab et al., 2006), paramagnetism-based structural restraints (Arnesano et al., 2005; Cheng and Markley, 1995; Clore and Iwahara, 2009; Nitsche and Otting, 2017; Orton and Otting, 2018), ab-initio calculations to map the electron delocalization onto the surrounding ligands (Machonkin et al., 2005), protocols to minimize the blind sphere around the cluster (Banci et al., 2013; Banci et al., 2014), and the obtainment of PRE-only NMR structures (Trindade et al., 2020).

In the new millennium, microbiologists, cell and molecular biologists, and eventually geneticists entered into the scenario, affording the study of the pathways for the cluster biosynthesis in Fe–S proteins in model organisms and in humans thus moving the frontier in Fe–S protein research toward a system-wide perspective (Lill, 2009; Rouault and Tong, 2008; Schmucker and Puccio, 2010). Within this context, where cell biology, integrated structural biology, metalloproteomics, and spectroscopy form a unique research platform that provides a molecular view of Fe–S protein assembly processes and trafficking pathways, paramagnetic NMR contributes to characterize proteins involved into the Fe–S assembly machineries.

## 2 MitoNEET: (another) protein in search of a function?

In order to discuss how "old fashion" NMR spectroscopy of paramagnetic systems can contribute to tackle challenging aspects of Fe–S proteins functions and how effective NMR can be as a fingerprint technique for the characterization of Fe–S proteins, we here analyze the case of the mitoNEET protein. The human CDGSH Fe–S domain-containing protein (also known as mitoNEET) is the first identified member of a novel family of Fe–S proteins, named "NEET" proteins, due to the presence of the C-terminal amino acid sequence Asn-Glu-Glu-Thr (NEET) (Colca et al., 2004). MitoNEET is an integral outer mitochondrial membrane (OMM) protein of ~ 12 kDa and is characterized by unique fold and cluster-binding mode. The protein is anchored to the OMM through an N-terminal transmembrane domain (residues 14-32) (Wiley et al., 2007) while the soluble part points toward the cytosol and is composed of two main domains, as showed by X-ray crystallography (Baxter et al., 2011; Hou et al., 2007; Lin et al., 2007; Paddock et al., 2007): a β-cap domain, and a CDGSH cluster binding domain of 39 aa, containing the highly conserved CXCX$_2$(S/T)X$_3$PXCDG(S/A/T)H motif. The protein dimerizes through the formation of a three-stranded sheet involving residues 56-61 of one monomer and residues 68-71 and 101-104 of the second monomer (Hou et al., 2007; Lin et al., 2007; Paddock et al., 2007). The dimer interface is further stabilized by an intermolecular hydrogen bond between His-58 and Arg-73 (Paddock et al., 2007), and by two symmetric hydrophobic cores, comprising Ile-45, Ile-56, Trp-75, Phe-80 of one monomer, and Val-98 of the second monomer (Lin et al., 2007). Each subunit of the dimer binds one [Fe$_2$S$_2$]$^{2+/+}$ cluster with an unprecedented set of ligands, formed by three cysteines (Cys-72, Cys-74, Cys-83) and one histidine

(His-87), being different from the all-cysteine ligand motif found in ferredoxins and from the two cysteines and two histidines motif found in Rieske proteins.

The protein has been linked to different cellular processes and human pathologies. It has been shown that mitoNEET plays a key role in the regulation of iron and of the reactive oxygen species (ROS) homeostasis in cells (Kusminski et al., 2012), and in the modulation of mitochondrial bioenergetics, by regulating lipid and glucose metabolism (Kusminski et al., 2012; Vernay et al., 2017; Yonutas et al., 2020). In addition, MitoNEET is overexpressed in human epithelial breast cancer cells, where it maintains the mitochondrial functions and avoids the accumulation of iron and ROS in mitochondria (Salem et al., 2012; Sohn

et al., 2013). MitoNEET also plays a role in obesity, promoting lipid accumulation in adipocytes while preserving insulin sensitivity (Kusminski et al., 2012, 2014; Moreno-Navarrete et al., 2016), and in neurodegeneration (Geldenhuys et al., 2017). Moreover, mitoNEET was also found to be the primary mitochondrial target of the thiazolidinedione class of insulin-sensitizing drugs (TZDs) such as the antidiabetic drug pioglitazone (Colca et al., 2004), although the role that mitoNEET plays in the etiology of type 2 diabetes and in mediating some of the effects of TZDs remains to be determined.

The mechanisms by which mitoNEET participates in the aforementioned cellular processes are still elusive. However, it has been proposed that, for most of the cellular functions, the $[Fe_2S_2]$ clusters of dimeric mitoNEET might play a crucial role, possibly acting as redox- or pH-sensors for mitochondrial functions, and/or being transferred to cytosolic apo proteins in response to the redox states of the cells (Ferecatu et al., 2014; Lipper et al., 2015; Zuris et al., 2011). Indeed, mitoNEET redox state can be regulated *in vitro* by biological thiols such as reduced glutathione (GSH), L-cysteine, and *N*-acetyl-L-cysteine

(Landry and Ding, 2014) , reduced flavin nucleotides (Landry et al., 2017; Wang et al., 2017; Tasnim et al., 2020) and proteins such as human glutathione reductase (Landry et al., 2015) and human anamorsin (Camponeschi et al., 2017). On the other hand, mitoNEET is also able to repair Fe–S proteins, by reloading Fe–S clusters onto cytosolic proteins whose Fe–S clusters have been removed or altered (Ferecatu et al., 2014), although how the cluster of mitoNEET is regenerated after the transfer it is still elusive. The electronic properties and chemical reactivity of mitoNEET clusters have been extensively investigated so

far through several biophysical and biochemical techniques. Two-dimensional standard NMR and circular dichroism (CD) spectra acquired on the mitoNEET C-terminal cytosolic domain (res 44-108) showed that the unique fold adopted by each subunit in the holo protein is strictly related to the presence of a cluster, that can be disassembled and reassembled *in vitro*, inducing, respectively, unfolding and refolding of the protein (Ferecatu et al., 2014). EPR spectroscopy, performed on *E.coli* cells containing the overexpressed cytosolic domain of human mitoNEET, showed that in the cytoplasmic cellular environment

the two $[Fe_2S_2]$ clusters are in the reduced state (Landry and Ding, 2014), as expected given their ~0 mV midpoint redox potential, measured *in vitro* at pH 7.5 (Bak et al., 2009; Tirrell et al., 2009). The oxidation state of the $[Fe_2S_2]$ clusters of mitoNEET plays a crucial role in the *in vitro* cluster transfer activity of the protein, since only $[Fe_2S_2]^{2+}$ and not $[Fe_2S_2]^+$ clusters transfer from holo mitoNEET to apo recipient proteins have been observed (Ferecatu et al., 2014; Lipper et al., 2015; Zuris et al., 2011). This has led to the definition of "active state" and "dormant state" for, respectively, the oxidized and reduced

state of the cluster (Golinelli-Cohen et al., 2016). The stability of mitoNEET clusters can also be tuned by several other factors. The presence of a histidine residue in the first coordination sphere makes mitoNEET $[Fe_2S_2]$ clusters pH-sensitive: it was

observed that, below pH 6.0, the His-87 ligand is protonated and the release of the clusters in solution or their transfer to apo recipient proteins *in vitro* is facilitated (Ferecatu et al., 2014; Golinelli-Cohen et al., 2016; Lipper et al., 2015; Zuris et al., 2011). Moreover, UV-visible and one-dimensional NMR spectroscopic studies showed that the interaction of mitoNEET with the antidiabetic drug pioglitazone increases the [$Fe_2S_2$] clusters stability by a factor of ≈10 with respect to a control sample lacking pioglitazone (Paddock et al., 2007). The same studies showed that pioglitazone causes perturbations in the overall protein structure, and in particular it affects the resonances of aromatic residues (Trp or Phe), although they were not residue-specifically identified. On the other hand, it has been shown through NMR spectroscopy that, upon interaction of mitoNEET with reduced Nicotinamide Adenine Dinucleotide Phosphate (NADPH), the Fe–S clusters are destabilized, and the protein undergoes unfolding (Zhou et al., 2010).

Despite the number of studies and techniques applied for the investigation of the mitoNEET function, an atomic level characterization in solution is still missing. NMR spectroscopy is usually the election technique when it comes to atomic level characterization and to structural investigation of protein-protein or protein-ligand interactions in solution. However, the attempts of investigating mitoNEET through standard NMR spectroscopy failed due to severe line broadening caused by the presence of the paramagnetic clusters (Paddock et al., 2007; Zhou et al., 2010). We show here how paramagnetic NMR provides a powerful fingerprint of the cluster environment, able to provide residue-specific information. A protocol is here proposed in order to conjugate spectroscopic information arising from "classical" paramagnetic NMR with double and triple resonance experiments customized with respect to the relaxation properties of the specific systems to be studied. This protocol can be then combined with "classical" biomolecular NMR experiments, in order to overall detect a larger number of signals and extend the ability of NMR to map protein-protein and protein-ligand interactions.

# 3 Materials and Methods

## 3.1 Cloning, overexpression and purification of mitoNEET

The cDNA coding for the cytoplasmic domain (residues 32-108) of human mitoNEET (UniProtKB/Swiss-Prot: Q9NZ45) was acquired from Eurofins Genomics. The gene was amplified by PCR and directionally cloned into the pET151-D/TOPO vector (Invitrogen), which add a 6xHis tag followed by a TEV cleavage site and an additional GIDPFM aminoacidic sequence at the N-terminus of the protein. Rosetta 2(DE3) competent *E. coli* cells (Stratagene, La Jolla, CA) were transformed with the obtained plasmid, and were grown in Luria Bertani (LB) or M9 minimal medium (supplemented with 1 g of ($^{15}NH_4$)$_2SO_4$ and 3 g of $^{13}$C-glucose per liter), containing 1 mM ampicillin and 1 mM chloramphenicol at 37 °C under vigorous shaking, up to a cell $OD_{600}$ of 0.8. Expression of the holo form of mitoNEET was induced by adding 0.4 mM Isopropyl β-D-1-thiogalactopyranoside (IPTG) and 400 μM $FeCl_3$. Cells were grown at 25 °C overnight and harvested by centrifugation at

7500g. The cell pellet was resuspended in 20 mM Tris-HCl buffer pH 8.0, 500 mM NaCl, 5 mM imidazole, 0.01 mg/ml DNAase, 0.01 mg/ml lysozyme, 20 mM MgSO$_4$ and 5 mM DTT and lysed by sonication. All the following purification steps were performed in anaerobic conditions (O$_2$ < 1 ppm), inside an inert gas (N$_2$) glove box workstation (LABstar, MBRAUN). The clarified supernatant was loaded onto a HiTrap chelating HP column (GE Healthcare) and the protein was eluted with 20 mM Tris-HCl, pH 8.0, 500 mM NaCl, 500 mM imidazole. Cleavage of the tag was achieved by incubation with 6xHis tagged TEV protease in cleavage buffer (20 mM Tris-HCl pH 8.0, 5 mM imidazole, 500 mM NaCl, 0.3 mM EDTA, 3 mM DTT) overnight at room temperature. The solution was loaded onto a HisTrap FF column (GE Healthcare) to separate the protein from the cleaved 6xHis tag and from 6xHis tagged TEV protease. The oxidized form of [Fe$_2$S$_2$]-mitoNEET was obtained by adding up to 5 mM K$_4$Fe(CN)$_6$ to the protein solution. The reduced form of [Fe$_2$S$_2$]-mitoNEET was obtained by adding up to 5 mM sodium dithionite to the protein solution. The protein was then buffer-exchanged using a PD10 desalting column in degassed 50 mM phosphate buffer pH 7.0 containing 10% (v/v) or 100% D$_2$O for NMR experiments. UV-visible spectra of the oxidized and reduced form of mitoNEET (figure A1 in Appendix A) were anaerobically acquired on a Cary 50 Eclipse spectrophotometer. Protein concentration was 40 μM.

### 3.2 NMR spectroscopy

#### 3.2.1 Diamagnetic experiments on [Fe$_2$S$_2$]-mitoNEET oxidized and reduced

All NMR experiments used for resonance assignment for either oxidized or reduced mitoNEET were recorded on a Bruker AVANCE 500 MHz spectrometer on 0.5 mM $^{13}$C,$^{15}$N-labeled samples in 50 mM phosphate buffer, pH 7.0, containing 10% (v/v) D$_2$O. All NMR spectra were collected at 298 K, processed using the standard Bruker software (Topspin) and analyzed through the CARA program. The $^1$H, $^{13}$C and $^{15}$N resonance assignment for both redox states were obtained through acquisition and analysis of HNCA, HNCO, HN(CA)CO, CBCA(CO)NH and HNCACB experiments. All experiments were collected using a 1 s recycle delay and 16 scans each fid, a part for HNCO which was recorded with 8 scans per point. Tridimensional time domain points were as follows: HNCA 1024 x 48 x 112 (16 ppm x 32 ppm x 36 ppm); HNCO and HN(CA)CO 1024 x 48 x 80 (16 ppm x 32 ppm x 17 ppm); CBCA(CO)NH and HNCACB 1024 x 48 x 128 (13 ppm x 32 ppm x 76 ppm). For the reduced form, also 3D experiments for side chain assignments were performed. HBHA(CO)HN and (H)CCH-TOCSY were collected, with a 1024 x 48 x 128 (14 ppm x 32 ppm x 14 ppm) and 1024 x 64 x 200 (16 ppm x 75 ppm x 75 ppm) data point matrices, respectively. For the (H)CCH-TOCSY, spin lock and recycle delays were about 16.3 ms and 1.2 s, respectively. Heteronuclear relaxation experiments on $^{15}$N-labeled samples for oxidized and reduced mitoNEET were collected at 500 MHz in order to measure the $^{15}$N backbone longitudinal (R$_1$) and transverse (R$_2$) relaxation rates, as well as the heteronuclear $^{15}$N[$^1$H] NOEs, and to obtain information on the quaternary structure of the protein. Chemical shifts differences between the two oxidation states of mitoNEET have been calculated using the following equation $\Delta_{HN} = ((\delta_H)^2 + (\delta_N/5)^2)^{1/2}$. Chemical shift data

of reduced and oxidized mitoNEET have been deposited in the BMRB database with accession number 50681 and 50682, respectively. Residues experiencing largest chemical shift differences are shown in figure A2 in Appendix A.

### 3.2.2 Paramagnetic tailored experiments on [Fe$_2$S$_2$]-mitoNEET reduced and oxidized

MitoNEET is paramagnetic since it binds a [Fe$_2$S$_2$] cluster. Therefore, its relaxation and chemical properties in both oxidation states are strongly affected by the cluster and a full characterization requires paramagnetic tailored experiments. Proton detected 1D experiments were performed at temperatures ranging from 283 K to 298 K on both oxidized and reduced forms. Spectra were recorded on a Bruker AV600 MHz spectrometer, equipped with a 5 mm, $^1$H selective high-power probe without gradients. Experiments were performed with a standard water presaturation pulse, with acquisition and recycle delays of 85 ms and 230 ms, respectively. The length of 90° pulse was ~ 7 μs. 32k scans were collected, using a dwell time of 2.6 μs and analog filter mode. A 60 Hz line broadening filter was used prior to Fourier Transformation. Proton 1D NOE experiments were collected on the hyperfine shifted signals of the oxidized form of the protein at 283 K on a Bruker AV600 MHz spectrometer. NOE experiments were collected in an interleaved way, following a well-established methodology (Banci et al., 1989). Selective on- and off-resonance saturation was applied during the inversion recovery delay of a superWEFT experiment, recorded with 80 ms and 105 ms of inversion and recycle delays, respectively. Typically, selective saturation was kept for 75 ms during the inversion recovery period. On-resonance experiments were obtained by suppressing about 60 % of signal intensity. Off-resonance experiments were obtained by irradiating, using the same power of the on-resonance experiment, at ($\omega_{on} \pm \omega_{offset}$) frequencies. For each 1D NOE experiment, different $\omega_{offset}$ values were chosen: 1.6 kHz for signal A, 2.1 kHz for signal D and 1.8 kHz for signal E. Experiments have been acquired with ~ 900k, 450k and 350k scans, for signals A, D and E respectively. In order to analyze the 1D difference spectra in the diamagnetic region, a 10 Hz line broadening filter was applied prior to Fourier transformation. In order to optimize the detection of HN signals close to the Fe–S cluster and experiencing paramagnetic relaxation enhancement, the IR-$^{15}$N-HSQC-AP was used (Ciofi-Baffoni et al., 2014). The IR-HSQC-AP experiments were collected using a Bruker AVII 700 MHz spectrometer, equipped with a 5 mm TXI probe. The experiments were collected with 4096 scans over a 512 x 80 data point matrix, using 16.5 ms and 13.7 ms as acquisition delays in the direct and indirect dimensions, respectively. Between the 180° and the 90° $^1$H pulses of the Inversion Recovery block, an inter-pulse delay of 18 ms was used, while the recycle delay following the acquisition time was 11 ms. An INEPT transfer delay of 833 μs was used. These parameters will significantly suppress the intensity of signals with $^1$H$_N$ R$_1$< 20 s$^{-1}$, providing positive peaks for $^1$H$_N$ R$_1$ < 40 s$^{-1}$ and negative peaks for $^1$H$_N$ R$_1$ > 40 s$^{-1}$. $^{13}$C direct detected CON experiments on the reduced state of mitoNEET (Mori et al., 2010), were acquired on a Bruker AVII 700 MHz spectrometer, equipped with a TXO probe, to identify and assign backbone C$_{(i-1)}$/N$_{(i)}$ connectivities. The diamagnetic version of the experiment was acquired with 64 scans over a 1024 x 256 data point matrix, using 58 ms and 31 ms as acquisition delays in the direct and indirect dimensions, respectively. A recycle delay of 2.5 s was used, together with a 12.5 ms delay for the C'/N INEPT transfer. In order to suppress the $^1J$ C'-Cα, the virtual decoupling was achieved via IPAP. This approach has been reported very effectively for $^{13}$C homodecoupling

and heteronuclear decoupling (Andersson et al., 1998; Bermel et al., 2006; Ottiger et al., 1998). The paramagnetic tailored experiment was recorded using the same pulse sequence of the diamagnetic CON experiment and acquisition parameters were optimized for the identification of fast relaxing signals. The tailored experiment was acquired with 2048 scans over a 400 x 160 data point matrix, using 31 ms and 22 ms as acquisition delays in the direct and indirect dimension, respectively. Recycle delay and C'/N INEPT transfer length were taken as short as 200 ms and 8 ms, respectively. The short recycle delay was used to enhance signal intensity of peaks with $^{13}$C' $R_1 > 5$ s$^{-1}$. The C'/N INEPT transfer was shortened from 12.5 ms to 8 ms to incorporate the IPAP module and account for fast relaxing signals affected by paramagnetic clusters. The 8 ms transfer delay provides a slightly lower efficiency of the IPAP $^1J$ C'-C$\alpha$ decoupling and gives rise, in principle, to incomplete suppression of doublet components. However, as we are dealing with broad signals, the effect is hidden by paramagnetic broadening. The efficiency of C'-N coherence transfer vs $R_2$ $^{13}$C' relaxation is reported in Appendix A, figure A3. The relevant data sets are available from https://doi.org/10.5281/zenodo.4442396 (Camponeschi et al., 2021).

# 4 Results

## 4.1 Sequence-specific assignment

Size-exclusion chromatography (data not shown) and heteronuclear $^{15}$N relaxation measurements optimized for diamagnetic systems showed that mitoNEET is anaerobically purified as a homodimer, as previously reported for similar mitoNEET constructs (Hou et al., 2007; Lin et al., 2007; Paddock et al., 2007). Indeed, relaxation data of the protein regions not affected by the paramagnetic center (Appendix A, figure A4) account for a molecular reorientational correlation time of $11.6 \pm 0.8$ ns, which is consistent with the 19.4 kDa molecular weight of dimeric mitoNEET (Mori et al., 2008; Rossi et al., 2010).

A series of double and triple resonance experiments, recorded using the conventional experimental set-up for diamagnetic proteins (Ab et al., 2006), achieved about 60% of the backbone assignment for both reduced and oxidized mitoNEET (BMRB code 50681 and 50682, respectively). This is in agreement with previous NMR studies (Golinelli-Cohen et al., 2016; Zhou et al., 2010) which reported similar percentages for mitoNEET backbone NMR assignments.

As shown in **Figure 1a**, the longest missing stretch in the assignment is the region Tyr-71-Asn-91, which encompasses the cluster binding residues (Cys-72, Cys-74, Cys-83 and His-87); further missing assignments are for residues located around the [Fe$_2$S$_2$] cluster. All together these not detected signals define a "blind sphere" around the cluster, due to paramagnetic relaxation enhancement (Arnesano et al., 2005; Battiste and Wagner, 2000; Donaldson et al., 2001; Otting, 2010).

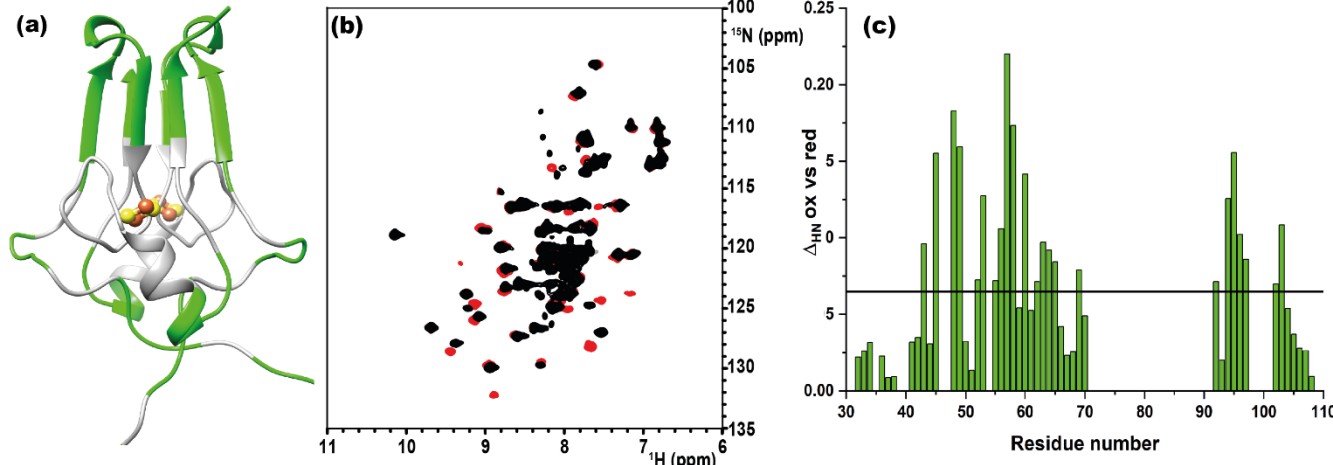

Figure 1. (a). Crystal structure of mitoNEET (2QD0). Protein segments in green could be identified in HSQC experiments and sequentially assigned in triple resonance experiments for both oxidation states. (b). $^1$H-$^{15}$N HSQC overlap of mitoNEET oxidized (red) and reduced (black) at 700 MHz at 298K. (c). Chemical shifts differences between the two oxidation states of the protein. The black bar is the average plus one standard deviation, with the residues above it being those significantly different. The residue number follows the PDB X-rays crystal structure 2QD0 (Lin et al., 2007). .

The two oxidation states of mitoNEET showed different chemical shifts, as shown by the superimposition of their $^1$H-$^{15}$N HSQC spectra (**Figure 1b**). The observed differences (**Figure 1c**) are relatively small and are not determined by paramagnetic effects, because the contribution to chemical shifts for not coordinated residues is negligible. In electron transfer proteins, where chemical shift differences have been widely analyzed, redox shifts have been correlated to the electron-transfer process (Lehmann et al., 2002; Pochapsky et al., 2001; Xia et al., 1998). Here the observed changes (highlighted in figure A2, Appendix A) seem to affect mainly the protein regions involved in inter-subunit contacts, such as the network of interactions involving Asp-96 with Ile-45 or Phe-60 with Ile-103.

## 4.2 Paramagnetic NMR

Albeit paramagnetic relaxation prevents the sequence specific assignment of the region around the cluster, its first coordination sphere can be monitored via paramagnetic $^1$H NMR spectroscopy. While no hyperfine shifted signals were observed for the reduced [Fe$_2$S$_2$]$^+$-bound form of mitoNEET (data not shown), the 1D NMR spectrum of the oxidized [Fe$_2$S$_2$]$^{2+}$-bound form of the protein (**Figure 2**) showed five signals in the 60-20 ppm region, and five additional, much sharper, signals in the 15-10 ppm region. As reported in **Table 1**, the linewidths of the five signals labeled A-E, measured at 600 MHz, are in the range 1500-3000 Hz, while the signals in the 15-10 ppm region have linewidths between 70 and 250 Hz. All A-E signals showed antiCurie temperature dependence. When the spectrum was recorded on a sample in D$_2$O, signal B (at 46.8 ppm) significantly decreased its intensity (Appendix A, figure A5) thus indicating that it is due to an exchangeable H$_N$ proton. In the 15-10 ppm region all signals disappeared, except the peak at 10.6 ppm (labeled as F). This signal showed an antiCurie temperature

dependence and a linewidth >200 Hz; therefore, it is due to a proton experiencing hyperfine interaction with the cluster electron spin and thus belonging to the first coordination sphere of the cluster.

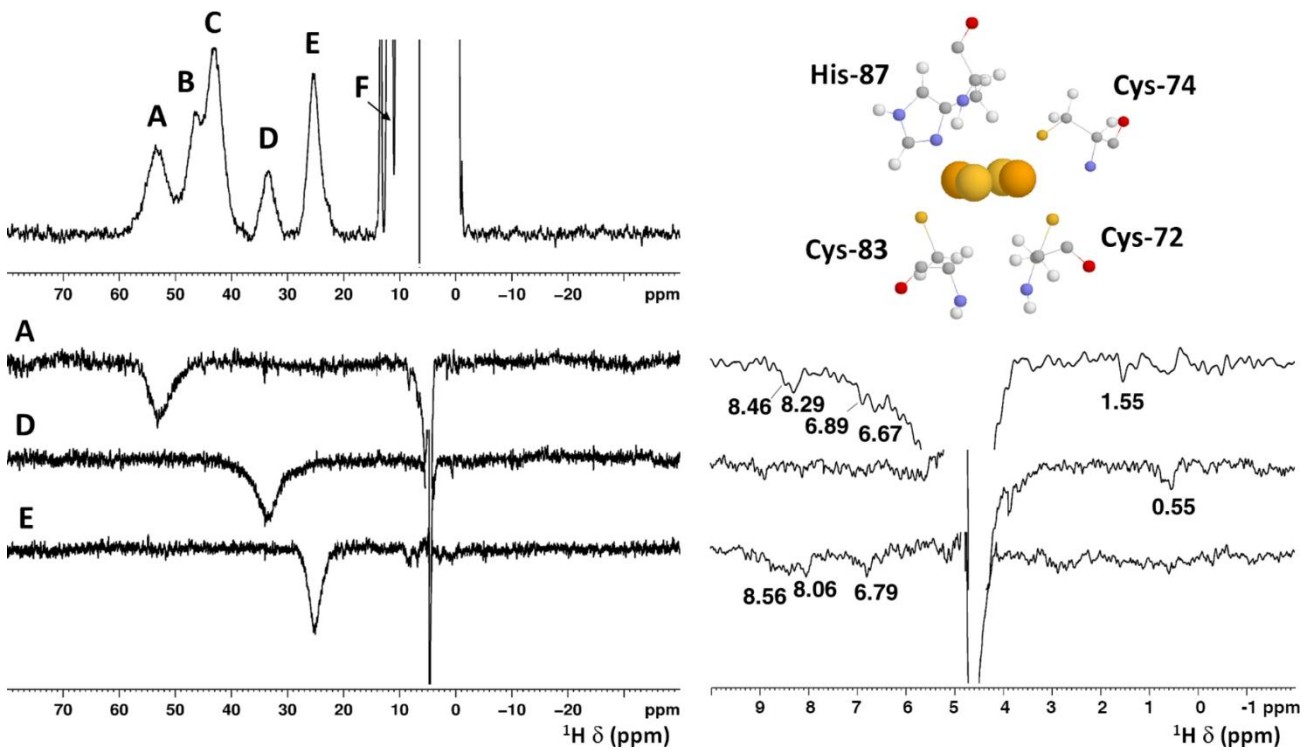

Figure 2: Upper panel: left - $^{1}$H NMR spectrum of oxidized [Fe$_2$S$_2$]$^{2+}$ mitoNEET, at 600 MHz, 283K; right - the cluster binding residues are shown (PDB ID: 2QD0 (Lin et al., 2007), protonated with UCSF Chimera). Lower panel: left - 1D NOE difference experiments on oxidized [Fe$_2$S$_2$]$^{2+}$ mitoNEET at 600 MHz, 283K. The letters indicate the signals which have been selectively irradiated to obtain the difference experiment; right - for each of the 1D NOE difference experiments reported on left panel, the 10-0 ppm region of the spectrum is shown. Peaks observed in the difference experiments are indicated by their chemical shifts.

Since the [Fe$_2$S$_2$]$^{2+}$ cluster has a negligible magnetic susceptibility anisotropy, pseudocontact contributions to the observed shifts can be neglected. Therefore, the downfield shifts observed for signals A-F are fully due to the contact contribution to chemical shift, thus suggesting that these signals originate from protons of cluster-bound residues. Indeed, their shifts, temperature dependences, and linewidths are fully consistent with protons belonging to residues bound to an oxidized [Fe$_2$S$_2$]$^{2+}$ cluster, with an electron spin ground state S=0 (Banci et al., 1990b). Therefore, the only possible assignment for signal B is the H$_N^{\varepsilon 2}$ of the iron bound His-87, which is the only exchangeable proton for which a sizable unpaired electron spin delocalization is expected.

**Table 1. Chemical shifts and linewidths of the paramagnetic $^1$H NMR spectrum of oxidized MitoNEET, recorded at 600 MHz, and the proposed signal assignment.**

| Signal | 283 K (ppm) | 293 K (ppm) | $\Delta\nu$ 293 K (Hz) | Proposed Assignment |
|---|---|---|---|---|
| A | 53.8 | 54.4 | 2700 | His-87 H$^{\delta 2}$ |
| B $_{exch}$ | 46.8 | 47.3 | 2500 | His-87 H$_N^{\varepsilon 2}$ |
| C | 43.4 | 43.9 | 2300 | Cys-83 H$^{\beta 3}$ |
| D | 34.1 | 34.3 | 1800 | Cys-74 H$^{\beta 2}$ |
| E | 25.7 | 25.9 | 1500 | Cys-72 H$^{\beta 3}$ |
| H$_{N\ exch}$ | 13.6 | 13.5 | 150 | |
| H$_{N\ exch}$ | 12.08 | 12.02 | 200 | |
| H$_{N\ exch}$ | 11.39 | 11.35 | 120 | |
| F | 10.52 | 10.61 | 250 | Cys-74 H$^{\alpha}$/His-87H$^{\beta 2}$ |
| H$_{N\ exch}$ | 10.15 | 10.15 | 70 | Glu-38 H$_N$ |


### 4.2.1 Proposed assignment of the paramagnetic $^1$H NMR spectrum.

The identification of the broad signal B as due to His-87 H$_N^{\varepsilon 2}$ opens the opportunity for a tentative assignment of the remaining paramagnetically shifted $^1$H NMR signals (**Figure 2**). Signals A-E have similar linewidths, corresponding to R$_2$ rates of ~

5000-10000 s$^{-1}$. The paramagnetic contribution to transverse nuclear relaxation rates arises from the sum of contact, dipolar and Curie spin terms (Bertini et al., 2017b). The shifts of signals A-E, together with ESEEM data on mitoNEET (Dicus et al., 2010) and NMR studies on other [Fe$_2$S$_2$]$^{2+}$ proteins (Trindade et al., 2021), suggest that hyperfine coupling constants in the range of A/h 1-3 MHz can be estimated for His imidazole ring and Cys $\beta$CH$_2$ protons. With these parameters, considering that mitoNEET is a dimer in solution, we expect a dominant contact contribution to transverse relaxation for meta-like imidazole

His protons, that are at about 5 Å from the nearest iron ion but a predominant dipolar contribution for protons that are 3-3.5 Å apart from the nearest iron ion. Indeed, the dipolar and the Curie spin terms are related to the metal-to-proton (MH) distance via a r$^{-6}_{MH}$ relationship (Solomon, 1955). Considering that signal B arises from a proton, His-87 H$_N^{\varepsilon 2}$, about 4.9 Å apart from the metal center and using the available X-ray structure (PDB ID: 2QD0, (Lin et al., 2007)) to obtain metal-to-proton distances, we can predict that signals arising from protons at 3.0-3.5 Å from the metal would experience linewidths > 4 kHz, being

therefore broadened beyond detection. The proposed scenario is summarized in **Table 2,** where the MH distances of the protons of cluster-bound residues are reported.

**Table 2. Distance-based proposed assignment of the paramagnetic NMR spectrum of oxidized MitoNEET. In the assignment columns, bold text indicates the proposed assignment according to 1D NOE experiments**

| Cys-72 Dist to $Fe_1$ Å | Assignment | Cys-74 Dist to $Fe_1$ Å | Assignment | Cys-83 Dist to $Fe_2$ Å | Assignment | His-87 Dist to $Fe_2$ Å | Assignment |
|---|---|---|---|---|---|---|---|
| $H_N$ 5.54 | HSQC-AP **8.61** | $H_N$ 3.54 | Beyond detection | $H_N$ 5.61 | HSQC-AP | $H_N$ 3.16 | Beyond detection |
| $H^\alpha$ 3.21 | Beyond detection | $H^\alpha$ 4.90 | **Signal F/n.o.** | $H^\alpha$ 3.39 | Beyond detection | $H^\alpha$ 4.99 | Not shifted |
| $H^{\beta2}$ 3.47 | Beyond detection | $H^{\beta2}$ 4.25 | Signal A-E **Signal D** | $H^{\beta2}$ 3.33 | Beyond detection | $H^{\beta2}$ 4.39 | **n.o./Signal F** |
| $H^{\beta3}$ 4.35 | Signal A-E **Signal E** | $H^{\beta3}$ 3.27 | Beyond detection | $H^{\beta3}$ 4.32 | Signal A-E **Signal C** | $H^{\beta3}$ 2.98 | Beyond detection |
| | | | | | | $H^{\varepsilon1}$ 3.06 | Beyond detection |
| | | | | | | $H_N^{\varepsilon2}$ 4.94 | Signal B |
| | | | | | | $H^{\delta2}$ 5.18 | Signal A-E **Signal A** |


According to these distances, once signals arising from protons at less than 4 Å from the metal are discarded because broadened beyond detection and signals from the aliphatic part of His-87 are discarded because arising from protons that are too far from the metal (in terms of chemical bonds) to experience A/h values larger than 1 MHz, the only possible assignments of signals

A, C, D, E are: Cys-72 $H^{\beta3}$, Cys-74 $H^{\beta2}$, Cys-83 $H^{\beta3}$ and His-87 $H^{\delta2}$ (reported in bold in **Table 2**). A specific assignment of these signals can be proposed using the 1D NOE difference experiments collected by saturating signals A, D and E, which could be selectively saturated (**Figure 2**). The selective saturation of signals with $T_2$ <0.2 ms is very difficult to accomplish (Banci et al., 1990a), as it requires too high power which gives poor selectivity and difference spectra with a low signal-to-noise ratio. Only very sparse and weak NOEs (less than 2%) could be measured from hyperfine shifted signals to peaks of the

diamagnetic region, consistent with the fact that NOE intensities are quenched for signals that experience paramagnetic relaxation enhancement. Even in the absence of hyperfine shift, transverse relaxation may provide significant line broadening of signals in the proximity of the cluster, thus making cumbersome the interpretation of the 1D difference spectra. The NOE difference experiments recorded upon the selective saturation of hyperfine shifted signals A, D and E (**Figure 2**) can be compared with the pattern of NOEs that, from each of the possible assignments, can be predicted on the basis of the X-ray

structure (PDB ID: 2QD0 (Lin et al., 2007)), protonated with UCSF Chimera (Pettersen et al., 2004). A more detailed description of the procedure is reported in Appendix A.

The proposed assignment is summarized in **Table 1**. Signals A-C, showing larger shifts and linewidths, are assigned to protons of residues bound to the $Fe_2$ of the cluster, while the less shifted and sharper signals D-E belong to residues bound to $Fe_1$. This

is an unprecedented feature, because in all $[Fe_2S_2]^{2+}$ cases investigated so far (Banci et al., 2013; Cai et al., 2017; Dugad et al., 1990; Skjeldal et al., 1991), the poor spectral resolution has prevented any attempt to analyze the electronic properties of the individual iron ions of the oxidized $[Fe_2S_2]^{2+}$ cluster. The two iron ions of the $[Fe_2S_2]^{2+}$ cluster in mitoNEET present different properties, that could arise either from a different electron spin relaxation time or from different spin delocalization mechanisms from the iron ions to the cluster-bound residues.

### 4.2.2 Paramagnetism-tailored HSQC experiments

The $^1$H-$^{15}$N-HSQC spectrum of the oxidized state of mitoNEET, recorded using standard conditions for diamagnetic systems, shows only 54 out of the 74 non-proline residues. Among them, 46 backbone signals were assigned using double and triple resonance experiments, while the remaining 8 $H_N$ resonances observed in the "diamagnetic" HSQC spectrum could not be sequentially assigned. Moreover, the protein construct contains also 6 additional vector-derived amino acids at the N-term site (see Materials and Methods), which have not been taken into account in this assignment. In the conventional, "diamagnetic" map, about 25% of the resonances (20 signals) remained unobserved, most likely due to paramagnetic broadening. From the X-ray structure, it appears indeed that 24 $H_N$ backbone protons are at less than 10 Å from one of the two iron ions. They belong to the 21 aa loop 70-91, encompassing the cluster binding region and to the small loop encompassing Pro-100 in the C-term part of the protein. However, customized modifications of the experimental setup of standard experiments such as $^1$H-$^{15}$N HSQC, allowed us to identify coherences otherwise undetectable in standard experiments. When the HSQC experiment is edited according to $^1$H $R_1$ relaxation properties, i.e. by adding an inversion recovery block before the $^1$H/$^{15}$N polarization transfer, the INEPT transfer is optimized to account for fast $^1$H $R_2$ relaxation (Ciofi-Baffoni et al., 2014), and a suitable choice of experimental parameters is used, as discussed in section 3.2.2, the intensity of HSQC peaks in the proximity of the metal center is enhanced. As a consequence, even before the sequence specific assignment is accomplished, it is possible to obtain a quite extended mapping of the signals of residues around the cluster which can be taken as a finger print of this protein region which usually constitutes the functional site. As shown in **Figure 3a**, when an IR-$^{15}$N-HSQC-AP experiment was performed, 11 additional $H_N$ signals (9 from backbone and 2 from side chain resonances), completely absent in conventional experiments, were observed. Furthermore, several $H_N$ signals, barely detectable or with very low intensity in the conventional experiment, significantly increased their intensity. The IR-$^{15}$N-HSQC-AP experiment retrieves therefore 9 amide resonances that are missing in the diamagnetic HSQC experiment. As it is well conceivable that the missing $H_N$ signals are due to the closest residues to one of the two metal ions, we can conclude that the IR-$^{15}$N-HSQC-AP experiment reduces the blind sphere around the cluster in oxidized mitoNEET, from ~ 10 Å to ~ 6.0 Å from the nearest iron ion.

For the reduced state of mitoNEET, the situation is slightly different: in the IR-$^{15}$N-HSQC-AP experiment only 6 additional backbone and 2 side chain $H_N$ signals were detected compared to the diamagnetic HSQC experiment, where 51 out of the 74 non-proline residues were detected, as shown in **Figure 3b**. Overall, 17 $H_N$ backbone resonances were missing in the IR-

HSQC-AP spectrum of the reduced $[Fe_2S_2]^+$ protein, setting the blind sphere around the cluster at about 8.0 Å, i.e. significantly larger than that observed in the oxidized form.

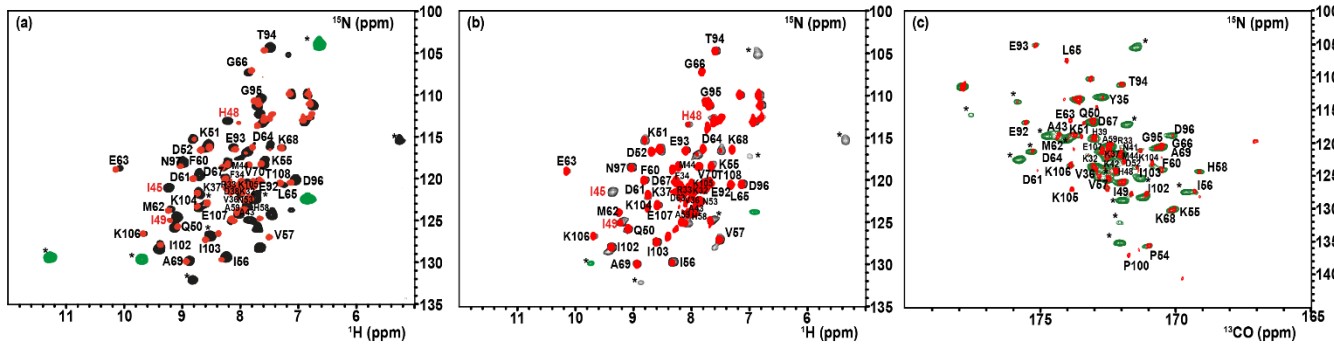


**Figure 3: Overlap of the diamagnetic HSQCs (red) and paramagnetic tailored $^{15}$N-IR-HSQC-AP (positive peaks: black/negative: green) for the oxidized (a) and reduced (b) forms of mitoNEET. (c) Overlap of diamagnetic (red) and paramagnetism tailored (green) CON experiments for the reduced state of mitoNEET. Peaks labeled with asterisks are those that are observed only in the paramagnetic-tailored experiments. Peaks observed but not assigned are labeled with X. Red-colored assignments refer to signals**
**that are only barely detectable in the diamagnetic experiments and could be sequentially assigned only in the paramagnetic-tailored experiments.**

### 4.2.3 $^{13}$C Detection experiments

$^{13}$C direct detection is nowadays a well-established experimental approach, particularly useful for paramagnetic systems (Arnesano et al., 2003; Bermel et al., 2006; Bertini et al., 2005; Kostic et al., 2002; Machonkin et al., 2002). The CON experiment is used as a protein fingerprint, complementary or alternative to $^{15}$N HSQC when the protein is unstructured (Ab et al., 2006; Brutscher et al., 2015; Contreras-Martos et al., 2017), or proline-rich or paramagnetic (Balayssac et al., 2006; Mori et al., 2010), like the present case. We used the reduced form of mitoNEET as a test system to assess the performances

of $^{13}$C detection. Unlike the previously discussed $^{15}$N HSQC experiment, the virtual decoupling of the $^1J$ Cα-C' with an IPAP scheme before $^{13}$C' direct detection does not allow the straightforward optimization of the CON experiment acquired in the antiphase mode. Therefore, we maintained the same CON pulse sequence used for the diamagnetic experiment and optimized the length of the C'/N INEPT delay according to relaxation-weighted transfer functions, as shown in figure A3 in Appendix A. For relaxation rates of the C'$_y$N$_z$ coherence faster than 10 s$^{-1}$, the efficiency of the transfer is significantly affected and the

decrease of the C'/N INEPT delay becomes mandatory. However, decreasing the delay below 9 ms give rise to incomplete decoupling of the $^1J$ Cα-C' doublet and a compromised set up has to be taken. In the CON experiment recorded under standard conditions we observed 49 C'-N correlations and 43 of them were assigned. When the experiment was optimized for paramagnetic systems, 13 additional C'-N correlations were observed (**Figure 3c**). Therefore, the use of a $^{13}$C direct detected experiment gives better results than the $^{15}$N IR HSQC-AP, in which only 6 paramagnetic H$_N$ peaks were observed because the

signal intensity is modulated by $^1H$ relaxation. The observed signals account for an estimated blind sphere of 6.5 Å from the nearest iron ion, smaller than that observed with the IR-HSQC-AP experiments.

## 5 Discussion

The NMR spectroscopy features of Fe–S proteins largely depend on the nature and properties of the bound Fe–S cluster(s). In any type of cluster, both $Fe^{3+}$ and $Fe^{2+}$ ions have a tetrahedral coordination and are in the high spin state. Only some combinations of iron oxidation states are present in proteins. For the $[Fe_2S_2]$ cluster, the cluster contains either two $Fe^{3+}$ ions (termed as the oxidized $[Fe_2S_2]^{2+}$ state), or it has one $Fe^{3+}$ and one $Fe^{2+}$ ion, in the so-called reduced $[Fe_2S_2]^+$ state. The extra electron of the reduced state can be either localized on a specific iron ion or it can be delocalized over the two iron ions, thus

being better described as $Fe^{2.5+}$ ions (or mixed valence iron ions). Furthermore, in both oxidation states, the iron ions are magnetically coupled. For $[Fe_4S_4]$ clusters, three oxidation states are possible: a $[Fe_4S_4]^{3+}$ state with three $Fe^{3+}$ and one $Fe^{2+}$ ions, a $[Fe_4S_4]^{2+}$ state, containing two $Fe^{3+}$ and two $Fe^{2+}$ ions and a $[Fe_4S_4]^+$ state, with one $Fe^{3+}$ and three $Fe^{2+}$ ions. The protein environment determines, for each $[Fe_4S_4]$ protein, the possible oxidation states of the cluster. Two different families of proteins are identified: High Potential Iron-Sulfur Proteins, that shuttle between the $[Fe_4S_4]^{3+}$ and the $[Fe_4S_4]^{2+}$ states, and ferredoxins,

that are stable in the $[Fe_4S_4]^{2+}$ and in the $[Fe_4S_4]^+$ states (Beinert et al., 1997; Bertini et al., 1995, 1997; Ciofi-Baffoni et al., 2018; Crack et al., 2012; Ollagnier-De Choudens et al., 2000; Rothery et al., 2004). As for the $[Fe_2S_2]$ cluster, the iron ions in a $[Fe_4S_4]$ cluster are magnetically coupled each other; depending on the coupling and on the electron distribution, each iron ion can be considered either as "purely" $Fe^{3+}$ or $Fe^{2+}$ ion or as a mixed valence $Fe^{2.5+}$ ion (Banci et al., 2018). In all cases, the paramagnetic centers are characterized by little, if any, magnetic susceptibility anisotropy. Therefore, a common feature of all

investigated Fe–S proteins is that the NMR hyperfine shifts are determined by the contact contribution and do not contain any through-space structural information. The contact shift depends on the electron spin ground state, on the hyperfine coupling constant (A/h) experienced by each nuclear spin, and on the magnetic coupling constant $J$ between pairs of iron ions. As we can see throughout a few examples, each type of cluster has a clear $^1H$ NMR fingerprint in each oxidation state, given by the proton signals of iron-bound Cys and His residues.

When, due to magnetic coupling, the electron spin ground state is S=0 and therefore the systems are EPR silent, such as the $[Fe_2S_2]^{2+}$ and $[Fe_4S_4]^{2+}$ clusters, paramagnetic NMR spectroscopy is crucial for identifying the type of cluster bound to the protein. For both $[Fe_2S_2]^{2+}$ and $[Fe_4S_4]^{2+}$ clusters, paramagnetism arises from excited electron spin states, populated at room temperature, and consequently the paramagnetic NMR shifts increase when raising temperature (Banci et al., 1990b; Bertini et al., 2017a). The observed hyperfine shifts discriminate efficiently between proteins containing $[Fe_4S_4]^{2+}$ or $[Fe_2S_2]^{2+}$ clusters.

In the $[Fe_4S_4]^{2+}$ case, contact shifts for cysteine βCH$_2$ signals are in the range 1-15 ppm (Bertini et al., 1992a) This is a highly conserved feature among $[Fe_4S_4]^{2+}$-containing proteins: hyperfine contact shifts have a Karplus-type dependence on the $\chi_2$

dihedral angle and, due to the relatively small line-width of the signals, they can be measured, assigned and converted into structural information (Bertini et al., 1994). In $[Fe_2S_2]^{2+}$-containing proteins, the contact contributions are about 2-4 times larger and the linewidths are about one order of magnitude larger than in $[Fe_4S_4]^{2+}$-containing proteins. This provides a clear

and unambiguous tool to discriminate among the two, EPR silent, $[Fe_4S_4]^{2+}$ and $[Fe_2S_2]^{2+}$ states. At variance with $[Fe_4S_4]^{2+}$ containing proteins, that always show very similar NMR spectra, different types of $[Fe_2S_2]^{2+}$ proteins provide different spectra, as summarized in **Figure 4**. For plant-type electron-transfer ferredoxins (Banci et al., 1990b) and for the Rieske-type ferredoxin from *Xanthobacter* strain Py2 (Holz et al., 1997), only a very broad and unresolved feature is observed, in the 28-35 ppm range, attributed as arising from the unresolved eight cysteine $\beta CH_2$ signals. On the contrary, other $[Fe_2S_2]^{2+}$ proteins, like

vertebrate ferredoxins (Skjeldal et al., 1991) and the human proteins ISCA1 and ISCA2 (Brancaccio et al., 2014; Banci et al., 2014) involved into the mitochondrial ISC machinery (Lill, 2009; Maio and Rouault, 2020), show a larger signals dispersion and, for human ferredoxins FDX1 and FDX2 (Cai et al., 2017; Machonkin et al., 2004; Xia et al., 2000), also larger chemical shifts, up to about 45 ppm. Although no individual resonance assignments have been proposed so far for any of these systems, the NMR spectra show line narrowing with respect to plant type ferredoxins. Other proteins, such as the mitochondrial protein

GLRX5 (Banci et al., 2014) and the Rieske component of Toluene 4-Monooxygenase (Xia et al., 1999) have signal linewidths similar to those of vertebrate ferredoxins but with smaller chemical shift values, i.e. between 20 and 30 ppm.

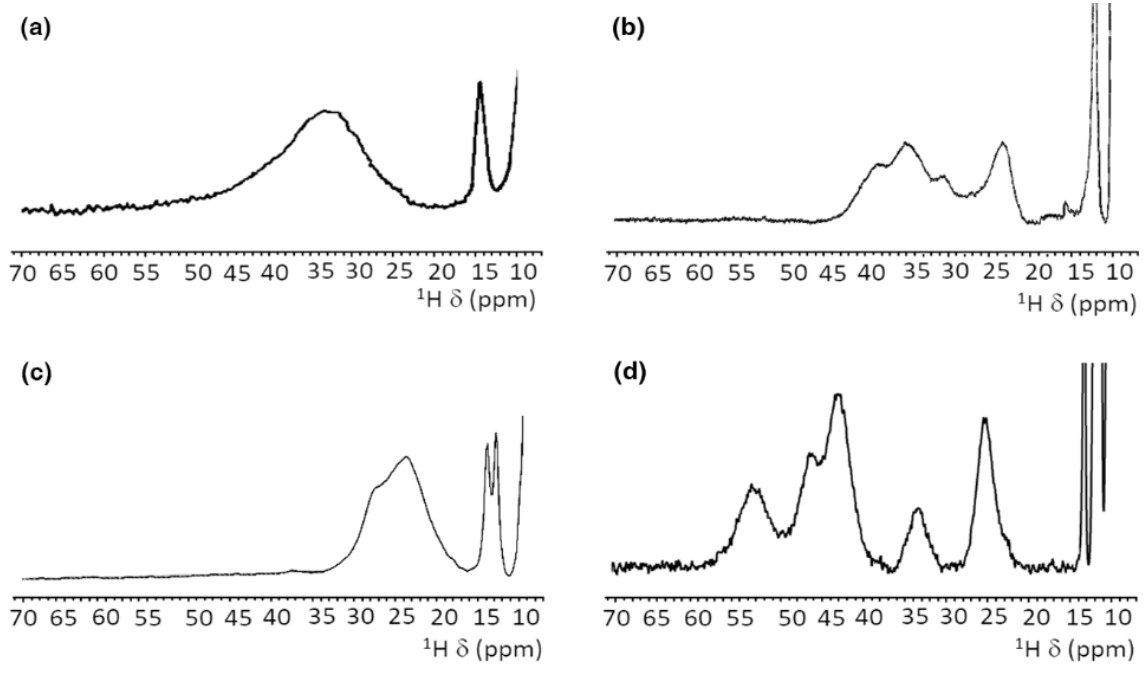

**Figure 4: Paramagnetic 1H NMR spectra of $[Fe_2S_2]^{2+}$ containing proteins: a) ferredoxin from red algae (Banci et al., 1990b); b)**
**human ISCA2 (Brancaccio et al., 2014); c) human Glutaredoxin-5 (Banci et al., 2014); d) mitoNEET.**

The paramagnetic NMR spectra of mitoNEET are significantly different from those reported for any of the aforementioned proteins and provide an additional contribution to the characterization of $[Fe_2S_2]^{2+}$-containing proteins, as shown in **Figure 4**. In oxidized, $[Fe_2S_2]^{2+}$-containing mitoNEET, the shifts of protons of cluster-bound residues are in the 60-25 ppm range. The

spreading of proton signals is, therefore, larger than in any of the previously investigated $[Fe_2S_2]^{2+}$ systems and the observed shifts are approximately 20% larger with respect to human ferredoxins FDX1 and FDX2 (Cai et al., 2017; Machonkin et al., 2004; Xia et al., 2000). This could be the consequence of a smaller antiferromagnetic coupling between the two iron ions which determines a larger population of the excited states of the electron spin energy ladder compared to other $[Fe_2S_2]^{2+}$ proteins. Spin polarization mechanisms on the histidine imidazolate ring (Bertini et al., 1992b; Ming and Valentine, 2014;

Spronk et al., 2018) may also provide a larger dispersion of the NMR signals of iron-bound histidine protons. Another feature, possibly contributing to the peculiar NMR spectrum of mitoNEET, is the coordination sphere of the cluster, which in mitoNEET is formed by three-Cys and one-His, thus breaking the symmetry of the typical four-Cys coordination of ferredoxins and other $[Fe_2S_2]$ cluster binding proteins. Interestingly, it has been shown that Cys-to-Ser mutations in Anabena-7120 ferredoxin increase the downfield shifts and signals dispersion (Cheng et al., 1994), supporting the proposal that a low

symmetry chromophore provides better resolved NMR spectra for the oxidized $[Fe_2S_2]^{2+}$ form.

On the other hand, the comparison between mitoNEET and Rieske proteins is not fully supporting the structural origin of the spectroscopic differences among the two classes of $[Fe_2S_2]$ proteins. In Rieske proteins, the $[Fe_2S_2]$ cluster is bound by two cysteine and two histidine residues. The two iron ions are highly inequivalent: the iron ion coordinated by two His residues is exposed to the protein surface while the iron ion coordinated by the two Cys residues is buried. This is similar to the cluster

environment in mitoNEET, in which the $[Fe_2S_2]$ cluster is bound by three cysteines and one histidine and the iron ion coordinated by Cys-83 and His-87 is close to the protein surface (Baxter et al., 2011; Hou et al., 2007; Lin et al., 2007; Paddock et al., 2007). Albeit mitoNEET and Rieske proteins share the feature of having non-equivalent iron sites and a mixed Cys-His coordination, the NMR spectra of their oxidized $[Fe_2S_2]^{2+}$ states are quite different. On the other hand, ESEEM experiments already showed that, in mitoNEET, the isotropic coupling constant of the iron bound histidine Nδ is larger than what observed

for the two iron-bound histidines in Rieske proteins, suggesting that small differences of iron coordination bonds and angles may affect the unpaired electron spin density delocalization onto the histidine ligand (Dicus et al., 2010). It is also likely that other structural features, such as the different metal binding motifs of the various $[Fe_2S_2]$ proteins, the composition of the second coordination sphere around the cluster and different networks of hydrogen bonds, that are known to play a crucial role in stabilizing the $[Fe_2S_2]$ cluster, are responsible for the specific properties of the paramagnetic NMR spectra.

When the reduced $[Fe_2S_2]^+$ state is considered, the differences among the various $[Fe_2S_2]$-binding proteins are even larger. Indeed, the different spectra observed for $[Fe_2S_2]^+$ ferredoxins have been associated with the different electronic properties of the cluster: when the extra electron is localized on one individual iron ion, relatively sharp and well separated NMR signals for all cysteine βCH₂ and αCH protons are observed. Indeed, the "valence localized" electron distribution provides much faster electron spin relaxation rates than the case of valence delocalized electrons. When valence is delocalized, the iron ions have much slower electron spin relaxation rates than in the localized valence pairs, thus determining much broader lines often


undetectable for $^1$H signals and eventually detectable, as very broad signals, only by $^2$H NMR measurements (Skjeldal et al., 1991; Xia et al., 2000). This model is also consistent with the NMR spectra of reduced $[Fe_2S_2]^+$ Rieske proteins, which show relatively sharp and well resolved NMR signals over a 100-20 ppm range. Instead, no signals are detected for reduced $[Fe_2S_2]^+$-mitoNEET, thus indicating that in mitoNEET the electron distribution within the cluster is different from Rieske proteins. A

similar behavior, i.e. the absence of detectable signals from Cys βCH$_2$ in the reduced $[Fe_2S_2]^+$ state, was observed for ISCA1 and ISCA2 (Brancaccio et al., 2014; Banci et al., 2014). Actually, Rieske proteins and the plant type ferredoxins (which share the same NMR features) act as electron transfer proteins, while ISCA1 and ISCA2 are involved into the assembly and transfer of the cluster. MitoNEET is supposed to play a major role in restoring the Fe-S cluster on cytosolic apo aconitase IRP1 in oxidative stress conditions (Ferecatu et al., 2014), and acts as a cluster transfer protein for several apo recipient proteins

(Ferecatu et al., 2014; Lipper et al., 2015; Zuris et al., 2011), both functions being based on a redox switch, activated by several cellular cofactors (Camponeschi et al., 2017; Landry et al., 2015, 2017; Landry and Ding, 2014; Tasnim et al., 2020). These findings are intriguing: a different coordination structure of the cluster, which determines the valence localization/delocalization within the cluster may be the origin of its different electronic properties, thus determining different NMR features, and possibly different functional properties. Specifically, for $[Fe_2S_2]^+$ clusters involved in electron transfer

processes the valence localization on an individual iron ion possibly makes the extra electron prone to be transferred to the redox partner. On the other hand, for $[Fe_2S_2]$ proteins involved into cluster transfer/assembly processes the two iron sites do not need to be inequivalent, while solvent accessibility might be in this case the driving factor for the cluster transfer event. In this respect is also very interesting that the protein region affected by the redox state changes is the inter-subunit one, as shown in figure A2 in Appendix A. A working hypothesis for future studies might be that, in order to perform its function, mitoNEET

switches between different conformational states, with the redox state change being one of the ways of regulating these transitions. Indeed, when mitoNEET passes from the "inactive", reduced state to the "active", oxidized state it adopts a less tight conformation that facilitates the cluster transfer to IRP1 or to other apo recipient proteins, possibly driven by higher solvent accessibility of the cluster itself.

**6 Conclusions**

The NMR characterization via one-dimensional paramagnetic NMR experiments offers insights into the electronic properties of the clusters, revealing features previously unobserved and unexpected. Indeed, it is another *tempo* of the tango-relationship between the electronic structure of Fe–S cluster and the biological functions of Fe–S proteins. The paramagnetic NMR

spectrum of oxidized mitoNEET has proton signals from cluster-bound residues characterized by linewidths sharper than any other $[Fe_2S_2]$ proteins characterized so far, while no signals are detected for the reduced form, at variance with the Rieske-type and plant-type ferredoxins.

The tailored design of double and triple resonance experiments using $R_1$ and/or $R_2$-based filters contributes to fill the gap between, on the one hand, the spectroscopic characterization of cluster and its first coordination sphere, obtained via 1D paramagnetic NMR and related methods and, on the other hand, the structural characterization of the protein regions unaffected by the hyperfine interaction. Even when the complete sequence specific assignment is not available, we can obtain relevant information on the peculiar active site of metalloproteins. For mitoNEET, our data indicate that paramagnetism-induced broadening is stronger in the reduced form of the protein; the electronic structure of the cluster is clearly one of the major changes when passing from the "inactive" reduced state to the "active" oxidized state, possibly highlighting the role of the electronic structure in driving functional properties of NEET proteins.

## Appendix A

**Additional Figures**

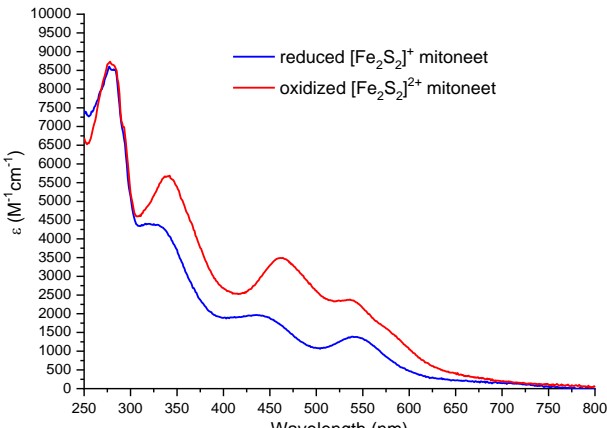

**Figure A1: UV-visible spectra of reduced [Fe$_2$S$_2$]$^+$- (blue line) and oxidized [Fe$_2$S$_2$]$^{2+}$- mitoNEET (red line). ε values are based on monomeric protein concentration (determined with Bradford assay).**

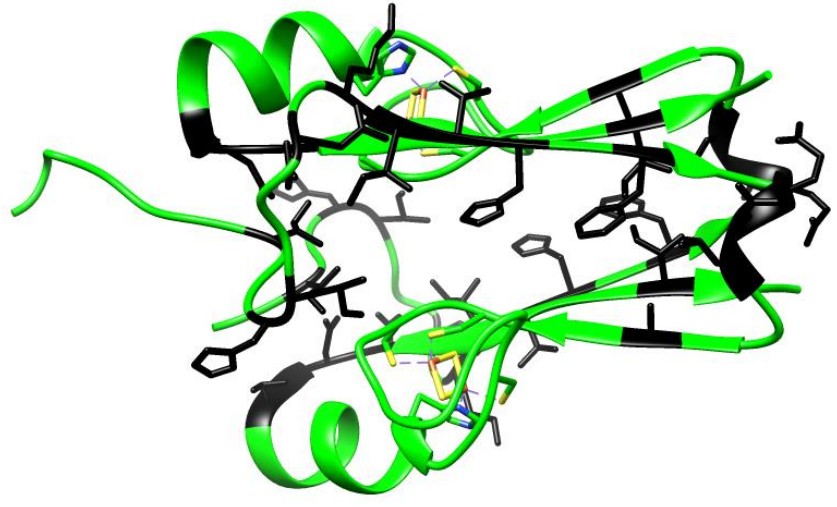


**Figure A2: Crystal structure of mitoNEET (2QD0). Residues experiencing the higher changes in the NH chemical shifts in the two different redox states as identified in the HSQC spectra are colored in black.**

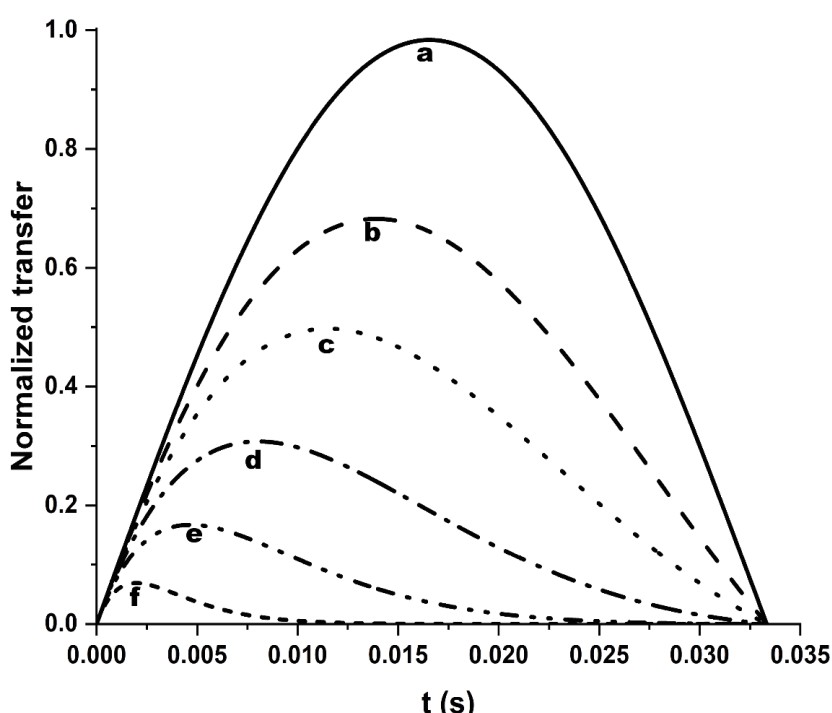


**Figure A3: Efficiency of an INEPT C'/N transfer function at different $^{13}$C' relaxation rates. We considered 15 Hz $J$ C'-N and different $R_2$ values: a no relaxation, b 25 s$^{-1}$, c 50 s$^{-1}$, d 100 s$^{-1}$, e 200 s$^{-1}$, f 500 s$^{-1}$. Letters have been drawn at the correspondence of the maximum values for each transfer function: b 13.9 ms, c 11.4 ms, d 8.0 ms, e 4.7 ms, f 2.0 ms. An $R_2$=25 s$^{-1}$ was taken as diamagnetic contribution to transverse relaxation (Charlier et al., 2016).**


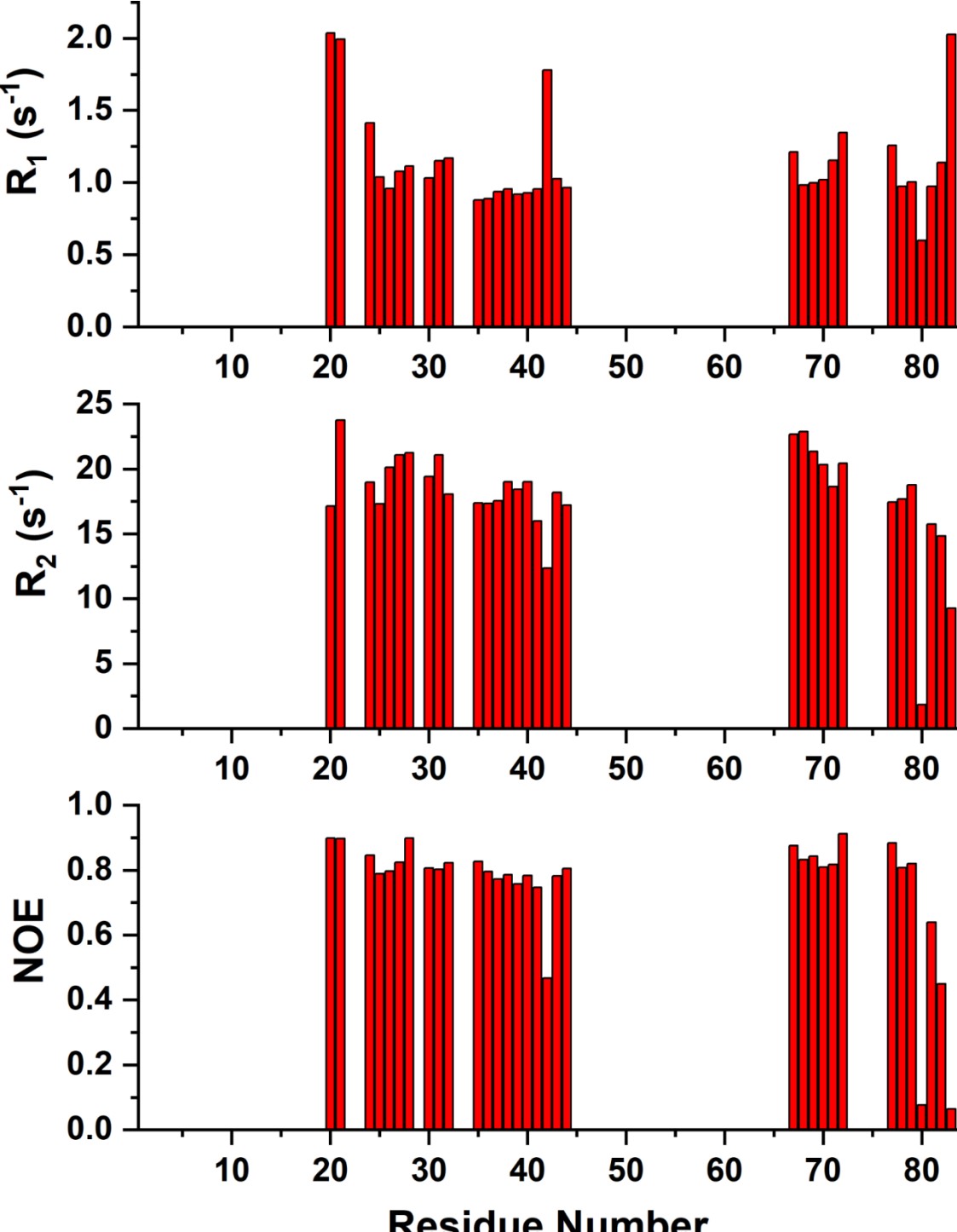

**Figure A4:** $^{15}N$ $R_1$, $R_2$, and $^{15}N\{^1H\}$ NOE values versus residue number of reduced mitoNEET obtained at 500 MHz and 298 K.


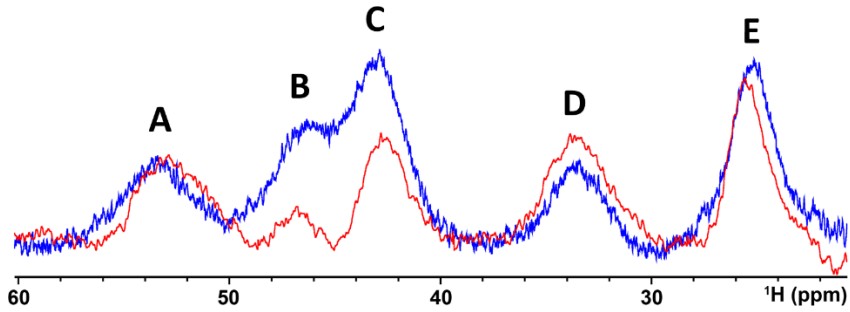

**Figure A5:** $^1H$ NMR spectrum of oxidized [Fe$_2$S$_2$]$^{2+}$ mitoNEET in 90 % H$_2$O and 10 % D$_2$O (blue line) and in 100 % D$_2$O (red line), at 600 MHz, 283K.


### Detailed description of the procedure used for the assignment of the paramagnetic $^1H$ NMR spectrum of oxidized mitoNEET

The selective saturation of signal A gives rise to five weak NOEs at 8.46 ppm, 8.29 ppm, 6.89 ppm, 6.67 ppm and 1.55 ppm. The NOE with the signal at 6.89 ppm is compatible with a possible NOE between His-87 $H_N^{\delta 2}$ (signal A) and Asn-91 $H_N^{\delta 2}$ (at

6.89 ppm) which is at 3.3 Å from His-87 $H_N^{\delta 2}$ and is relatively far from the two iron ions of the cluster (7.3 Å away from Fe$_2$ and 8.0 Å away from Fe$_1$) and therefore could be broad but detectable in the NOE experiment. An $H_N$ signal at 6.89 ppm arising from a side chain is observed in the $^{15}N$-HSQC-AP experiment. To support the hypothesis that signal A is due to His-87 $H^{\delta 2}$, we also observed that His-87 $H^{\delta 2}$, very close (2.3 Å) to Lys-55 $H^{\delta 2}$ (1.6 ppm from BMRB), experiences a NOE with the signal at 1.55 ppm. The observed NOE is relatively small, considering that the two protons are very close; however, Lys-

55 $H^{\delta}2$ is at 6.3 Å from the closest iron ion, and therefore it experiences paramagnetic relaxation enhancement.

Having assigned A as His-87 $H_N^{\delta 2}$, consequently signals C, D, and E are $H^{\beta}$ atoms of the three Cys residues that coordinate the cluster. Signal D shows a NOE at 0.55 ppm, which could be consistent with the proximity to a methyl group. The possibilities based on the X-ray structure (PDB ID: 2QD0 (Lin et al., 2007)) are either Cys-74 $H^{\beta 2}$ or Cys-83 $H^{\beta 3}$. Cys-74 $H^{\beta 2}$ is 2.7 Å apart from methyl groups of Val-98 CH$_3^{\gamma 2}$ (but at 5.4 Å from the closest iron and so it is broadened beyond detection)

and 2.8 Å apart from Ile-45 CH$_3^{\delta 1}$ (7.0 Å from the closest iron). Cys-83 $H^{\beta 3}$ is about 2.2 Å apart from Val-70 CH$_3^{\gamma 1}$ (5.3 Å from the closest iron and so it is broadened beyond detection). On this ground, we propose that signal D is assigned to Cys-74 $H^{\beta 2}$. Signal E gives a relatively strong NOE with a signal at 6.79 ppm, and NOEs with a broad signal at 8.56 ppm and with at least another signal at 8.06 ppm. Indeed, Cys-72 $H^{\beta 3}$ is at less than 3.0 Å from both $H^{\delta 1}$ and $H^{\epsilon 1}$ of Phe-80. Cys-72 $H^{\beta 3}$ is also close to several $H_N$ groups with distances to the cluster spanning from 3.5 Å to about 8 Å. In particular, we expect to observe

the intra-residue NOE between Cys-72 $H^{\beta 3}$ and Cys-72 NH, and an inter-residue NOE with Phe-82 HN, which is far from the cluster and therefore expected to be a sharp peak. Indeed, several NOEs of different linewidths in the amide region are observed

upon saturation of signal E, which can thus be assigned as Cys-72 $H^{\beta3}$. Signal C, which cannot be selectively irradiated, can then be assigned by exclusion as Cys-83 $H^{\beta3}$.

Once the strongly downfield shifted signals have been assigned, we can attempt the assignment of the non-exchangeable signal F, which experiences a contact contribution smaller than those of signals A-E. It would be consistent with either a Cys $H^{\alpha}$ proton (four $\sigma$ bonds apart from Fe ion) or a His $H^{\beta}$ (two $\sigma$ bonds away from the imidazole ring system, in which the electron is delocalized by spin polarization). According to the distances reported in Table 2 the two possible candidates are Cys-74 $H^{\alpha}$ and His-87 $H^{\beta2}$.

**Data availability.** Raw data are available at https://doi.org/10.5281/zenodo.4442396 (Camponeschi et al., 2021).

**Author contributions**. FC and AG conducted the experiments and the data analysis and wrote the paper; MP and LB planned the research, conducted data analysis and wrote the paper.

**Competing interests**. The authors declare that they have no conflict of interest.

**Acknowledgements**. The authors acknowledge the support of Instruct-ERIC, an ESFRI Landmark Research Infrastructure and specifically the use of the resources of the CERM/CIRMMP Italian Instruct Center.

**Financial support**. This research has been supported by Timb3, grant no. 810856, funded by the Horizon 2020 research and innovation programme of the European Commission, and by the Italian Ministry for University and Research (FOE funding) to the Italian Center (CERM/CIRMMP, University of Florence) of Instruct-ERIC.

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
