# Peer review of "The long-standing relationship between Paramagnetic NMR and Iron-Sulfur proteins: the mitoNEET example. An old method for new stories or the other way around?"

_Magnetic Resonance, 2021_

## Author Comment (AC2)

*In this study, Camponeschi et al use NMR to characterize mitoNEET, a mitochondrial $Fe_2S_2$ protein. By using 1D NOE experiments, $^{13}C$ direct-detected experiments, and the optimization of NMR experiments for paramagnetic systems, the authors show significantly reduction of the "blind" sphere of the protein around the paramagnetic cluster, thus allowing the detection of residues possibly involved in the biological function of mitoNEET. The study has significant implications in the fields of paramagnetic NMR and FeS proteins. Some revisions are recommended.*

We thank the reviewer for the comments and questions which allow us to better explain our work. We'll try to answer all the questions raised by the reviewer. Specifically:

*1. I have some general questions about the mitoNEET protein I hope the authors can help answer.*

*a) If mitoNEET can repair Fe-S proteins by donating its own $Fe_2S_2$ cluster, how does it reacquire the $Fe_2S_2$ cluster? Can the authors comment on the source of its $Fe_2S_2$ cluster?*

The source of mitoNEET cluster is still unknown. Ferecatu and coworkers (see Ferecatu et al., JBC, 2014, 289, 41, 28070-28086) demonstrated that the origin of iron and sulfur moieties required for mitoNEET maturation is mitochondrial, and that several components of the mitochondrial iron sulfur cluster (ISC) assembly and export machineries, such as ISCU, FXN, NFS1, HSC20, and ABCb7 are essential for the assembly of a $[Fe_2S_2]$ cluster on mitoNEET, whereas early and late acting components of the cytosolic iron sulfur cluster assembly (CIA) machinery are not. However, the mechanism of mitoNEET cluster maturation was not clarified, and, to the best of our knowledge, a specific protein able to repair mitoNEET cluster has not been identified yet.

*b) The redox states of mitoNEET are crucial for its function and stability. How are the redox states of mitoNEET regulated in cells?*

Although *in vivo* studies addressing how the redox states of mitoNEET are regulated in the cell are still missing, several *in vitro* studies showed that mitoNEET clusters can be reduced by many cellular reductants. Indeed, mitoNEET redox state can be regulated *in vitro* by biological thiols such as reduced glutathione (GSH), L-cysteine, and *N*-acetyl-L-cysteine (Landry AP, Ding H, *J Biol Chem* 2014, 289, 4307–4315), human glutathione reductase (Landry AP et al *Free Radic Biol Med.* 2015, 81, 119–127), reduced flavin nucleotides (Landry AP et al. *Free Radic Biol Med.* 2017, 102, 240–247; Tasnim H et al. *Free Radic Biol Med* 2020, 156, 11–19) and human anamorsin (Camponeschi F et al *JACS*, 2017, 139, 9479–9482), while NAD(P)H and NADH are not able to reduce mitoNEET clusters (Landry AP, Ding H *J Biol Chem* 2014, 289, 4307–4315). A comment on this aspect will be added to the manuscript.

*2. Some experimental details are needed.*

*a) For M9 media growth, how much $(^{15}NH_4)_2SO_4$ and $^{13}C$-glucose were supplemented?*

M9 media were supplemented with 1 g of $(^{15}NH_4)_2SO_4$ and 3 g $^{13}C$-glucose per liter. These details will be added in the Materials and Method section.

*b) What kind of anaerobic environment was used?*

The protein was purified and handled inside an inert gas glove box, working with $O_2 < 1$ ppm. This is now specified in the Materials and Methods section.

*c) Does the phosphate buffer contain any NaCl?*

No, it doesn't

*d) I assume there were additional steps to remove the extra $K_4Fe(CN)_6$ or sodium dithionite?*

$K_4Fe(CN)_6$/sodium dithionite were removed after oxidation/reduction of the cluster using a PD10 desalting column. This detail has been added to the Material and Methods section. Thanks for the comment.

*2) What's the $Fe_2S_2$: protein ratio 'as purified'? It would be helpful to include UV data to show the load of $Fe_2S_2$ on the protein in both redox states.*

Non-heme iron and acid-labile sulfide quantification data (not reported in the manuscript) obtained for anaerobically purified mitoNEET following a previously published procedure (Banci L. et al. *Chem. Biol.* 2011, *18*, 794–804), gave $2.0 \pm 0.1$ Fe/mitoNEET and $1.9 \pm 0.2$ $S^{2-}$/mitoNEET (mol/mol of monomeric protein; error is the standard deviation of 4 measurements), meaning that we purified mitoNEET with ~ one $[Fe_2S_2]$ cluster per monomer. UV-visible data are reported here for the reviewer and will be later included in the revised manuscript, as suggested by the reviewer. $\varepsilon$ values are based on monomeric protein concentration (determined with Bradford assay).

[Figure]

*3) The authors purified the protein in an anaerobic environment, I assume it's because the $Fe_2S_2$ is susceptible to oxidative damage. Would addition of 10mM $K_4Fe(CN)_6$ to the protein solution damage the $Fe_2S_2$ cluster?*

In order to avoid oxidation of the $[Fe_2S_2]^+$ clusters of mitoNEET or oxidative damage of the $[Fe_2S_2]^{2+}$ clusters of mitoNEET upon exposure to $O_2$, we worked in an anaerobic environment using an inert gas glove box. This ensured long term stability of mitoNEET $[Fe_2S_2]^{+/2+}$ clusters. Indeed, during the NMR experiments we didn't observe changes in the cluster-bound protons signals in the paramagnetic NMR experiments or changes in the HN amide backbone signals in the diamagnetic and paramagnetic $^1H$-$^{15}N$ experiments. Indeed, when the $[Fe_2S_2]$ cluster of mitoNEET is disassembled the protein undergoes a folded-unfolded conformational change and the HSQC spectrum of the protein changes significantly (Ferecatu et al., JBC, 2014, 289, 41, 28070-28086; Golinelli-Cohen et al. *J Biol Chem.* 2016, 291, 7583–7593). Such changes were not observed in the HSQC spectra of reduced or oxidized mitoNEET over a period of roughly 12 h, suggesting that the cluster is stably bound to the protein for all the NMR experimental time.

The same behavior was observed when 10 mM $K_4Fe(CN)_6$ was added to the protein solution and the removed by PD10. Indeed, it can be stated that damaging of the $[Fe_2S_2]$ cluster was not observed upon addition of $K_4Fe(CN)_6$.

*4) Is the purified mitoNEET protein a homodimer as shown in Fig. 1A?*

Yes, the protein was purified as a homodimer, as suggested by size exclusion chromatography data and by $^{15}N$ relaxation measurements. Indeed, the latter account for a $\tau_R$ value of $11.6 \pm 0.8$ ns, which is consistent with a dimeric state of the protein, whose molecular weight is ~18 kDa. The data will be added in appendix X in the revised version of the manuscript.

*5) In Fig. 1A, can the authors highlight the residues that are affected by different redox states?*

As suggested by the reviewer, we will modify figure 1 highlighting the residues affected by the different redox states. These residues belong to the inter-subunit region as pointed out also by the reviewer. Residues number involved in the redox switch are 45, 48, 49, 53, 55, 56, 57, 58, 60, 63, 64, 65, 69, 94, 95, 96, 97, 103. They are colored in black in the figure.

[Figure]

*6) Fig 1B, how were the chemical shift differences between two redox states calculated?*

The chemical shifts differences have been calculated using the following equation.

$\Delta_{HN} = ((\delta_H)^2 + (\delta_N/5)^2)^{1/2}$. In the revised version of the manuscript this will be included in the Materials and Method Section.

*7) It's intriguing to me that the redox state change would mainly affect the regions involved in inter-subunit contacts. Do the authors have any hypothesis why?*

We completely agree, it is very intriguing and interesting that the region affected by the redox state change is the inter-subunit one. Our hypothesis is that in order to perform its function mitoNEET has to switch between different conformational states, with the redox state change being one of the ways of regulating these transitions. Indeed, when mitoNEET passes from the "inactive", reduced state to the "active", oxidized state it might adopt a less tight conformation that facilitates the cluster transfer to IRP1 or to other apo recipient proteins, possibly driven by higher solvent accessibility of the cluster itself.

*8) There is no mention of Fig. 1C in the text. The author might add some.*

We will refer to Fig 1C in the manuscript according to the suggestion of the reviewer.

*9) Can the authors provide some explanations why no hyperfine shifted signals were observed for the reduced $[Fe_2S_2]^+$-bound form of mitoNEET?*

As reported in previous work (J Biol Inorg Chem. 2018; 23(4): 665–685), this a typical effect in mammalian $[Fe_2S_2]^+$, in particular in the case of the two irons ion pairs with delocalized valence.
This has been first described by J Markley and coworkers and interpreted as due to the fundamentally different patterns of electron delocalization observed, for reduced $[Fe_2S_2]^+$ centers in plant and vertebrate feredoxins (Skjeldal et al, Biochemistry. 1991; 30 (37), 9078-9083). When valence is delocalized, the iron ions have much slower electron spin relaxation rates than in the localized valence pairs, thus determining much broader lines often undetectable for $^1H$ signals and eventually detectable, as very broad signals, only by $^2H$ NMR measurements (Xia et al, Archives Biochem, Biophys, 2000, 373 (2), 328-334.)

*10) The authors should provide the data showing the broadening of signal B collected in $D_2O$.*

We report here for the reviewer the data showing the broadening of signal B in $D_2O$. The figure will be also added in the revised version of the manuscript.

[Figure]

*11) The authors might want to highlight the additional residues assigned by $^{15}N$-IR-HSQC-AP in the structure of mitoNEET.*

Actually, IR-HSQC-AP and CON experiments pointed out a number of resonances, unobserved in the diamagnetic experiments, that belong to the residues in the proximity of the cluster. However, the sequence specific assignment of these resonances, requires a quantitative analysis of $R_1$ and $R_2$ $H_N$ and $H_C$ rates, a series of triple resonance experiments optimized to provide scalar connectivities, $^{13}C$ paramagnetic HSQC data and an "a-la-carte" analysis in order to identify the scalar and dipolar connectivities to confirm the assignment. This is beyond the aim of this work.

*12) The labels in Fig.3 are too small to read, the authors might want to improve that.*

We will provide a figure with improved and bigger labels. Thanks for the comment.

---

## Author Response (AR1)

*Florence, March 5th, 2021*

*Professor Gottfried Otting*
*Editor of*
*Magnetic Resonance*

TITLE: The long-standing relationship between Paramagnetic NMR and Iron-Sulfur proteins: the mitoNEET example. An old method for new stories or the other way around?
AUTHORS: Francesca Camponeschi, Angelo Gallo, Mario Piccioli and Lucia Banci

Dear Gottfried,

We thank you and the reviewers for the active discussion and for the helpful comments. Herewith enclosed we are sending you the revised version of the manuscript, where we have addressed all points raised by reviewers.
In particular: - the section 3.2.2 of the article, dealing with the description of paramagnetic tailored experiments used throughout the manuscript, has been substantially rewritten; - the discussion has been expanded by couple of paragraphs; - Five Figures were added into the Appendix in order to better address some experimental and methodological aspects. Below please find the replies to each specific issue.

Anonymous Referee #1

*In this report Camponeschi et al. present NMR assignments of the dimeric membrane-anchored human CDGSH protein ("mitoNEET"). Each subunit contains a $Fe_2S_2$ cluster, and the current study aims at investigating its electronic properties in both the oxidized and the reduced states. Due to the paramagnetism of the cluster the resonance assignment requires separate sets of experiments for residues located outside a ~ 10 Å sphere and residues near the cluster. The former involves standard 3D backbone triple-resonance and side-chain experiments. In order to reduce the "blind sphere" around the cluster and observe very fast relaxing resonances 2D $^{15}N$-IR-HSQC-AP, developed a couple of years ago by one of the authors, and protonless $^{13}C$-detected CON experiments were carried out. Finally, a number of protons from Fe-coordinating residues were assigned using 1D NOE experiments in conjunction with X-ray structure derived distances. Conclusions about the electron distribution within the $Fe_2S_2$ cluster were drawn from the envelope of the hyperfine-shifted spectral region, which has some functional implications.*

**Prof Lucia Banci**
Centro Risonanze Magnetiche (CERM)
Via Luigi Sacconi, 6 – 50019 Sesto Fiorentino (Fi)
tel +39 055 4574273, e-mail: banci@cerm.unifi.it
www.cerm.unifi.it/about-us/people/lucia-banci

[Figure]

*Although I am not an expert for iron-sulfur proteins it appears to me that the system studied here is of high biological relevance (nice literature overview in the Introduction, by the way). Overall, the manuscript is very well-written, except for paragraph 3.2.2 (see below). It describes sound experimental work and comprehensible interpretation of the results. However, it mostly represents an application of established techniques to a well-studied protein. Considering its editorial policy, the current paper falls outside the scope of Magnetic Resonance.*

The manuscript describes the NMR characterization of the first coordination sphere of mitoNEET in its two oxidation states. Albeit a few NMR studies are available for the protein, the data reported here provide advancements with respect to the current knowledge on the protein. We believe that the mitoNEET case is a very nice example of how a protocol based on the combination of various experimental approaches tailored to paramagnetic systems spanning from the more recent IR-HSQC-AP to the "ancient" 1D NOEs, could provide insights into the knowledge of a challenging system of high biological interest

We also respectfully disagree with the reviewer that this work represents a characterization of a well-studied protein. The reviewer itself pointed out that electron distribution, unknown till now, drawn by the hyperfine shifts has some functional implication yet to be fully elucidated. We still believe that the NMR diamagnetic and paramagnetic characterization of this system might open up further and might allow a deeper characterization of mitoNEET's function which is still mostly elusive.

Furthermore, as a general comment regarding the appropriateness of this manuscript for Magnetic Resonance, we would like to stress that the indication we received was "the contributions should be related to Rob Kaptein's activities in the areas of spin hyperpolarization, spin chemistry, and biomolecular NMR". In this respect, we think that the application of NMR techniques tailored to paramagnetic proteins to improve our understanding of the active site of an intriguing FeS protein, whose understanding at the molecular level is far from being complete, is very respectful of Rob's activities.

Regarding the innovative technological aspects provided by this work, using mitoNEET as an example, we have developed here a general protocol that conjugates spectroscopic information arising from "classical" paramagnetic NMR with an extended mapping of the signals of residues around the cluster. The latter can be taken, even before the sequence specific assignment is accomplished, as a finger print of the protein region constituting the functional site of the protein.

We have now underlined this aspect and added few comments on this in the abstract (lines 22-26), in Section 2 (lines 140-144), in the results Section (lines 357-364; 392-398) and in the Conclusions (lines 520-524).

[Figure]

*Paragraph 3.2.2 ("Paramagnetic experiments on [Fe₂S₂]-mitoNEET reduced and oxidized") is to a large extent phrased in a lab-style language and should be rewritten. Some examples:*

*Heading: are the experiments paramagnetic or the samples?*

*Lines 190, 194, 200: "Spectra were recorded on a Bruker Av600 MHz" (spectrometer?)*

*Line 198: "Each experiment consisted from ~ 300k up to ~900k scans."*

*Line 202: "...using 16.5 ms and 13.7 ms as acquisition and a t1max delay...."*

*Line 209: "...2048 scans each fid were collected..."*

*Line 211: "...the IPAP approach was used for homodecoupling..." Does that mean virtual decoupling of $^1JC'Ca$?*

We substantially revised this paragraph and rephrased many sentences, taking into account all the reviewer suggestions.

- The paragraph heading was changed into "Paramagnetic tailored experiments on [Fe₂S₂]-mitoNEET reduced and oxidized"
- "Spectra were recorded on a Bruker AV600 MHz spectrometer, equipped with a 5 mm, 1H selective high-power probe without gradients."
- "Proton 1D NOE experiments were collected on hyperfine shifted signals of the oxidized form of the protein at 283 K on a Bruker AV600 MHz spectrometer."
- "Experiments have been acquired with ~ 900k, 450k and 350k scans, for signals A, D and E respectively."
- "The IR-HSQC-AP experiments were collected using a Bruker AVII 700 MHz spectrometer, equipped with a 5 mm TXI probe. The experiments were collected with 4096 scans over a 512 x 80 data point matrix, using 16.5 ms and 13.7 ms as acquisition delays in the direct and indirect dimensions, respectively."
- "$^{13}$C direct detected CON experiments on the reduced state of mitoNEET (Mori et al., 2010), were acquired on a Bruker AVII 700 MHz spectrometer, equipped with a TXO probe, to identify and assign backbone $C_{(i-1)}/N_{(i)}$ connectivities"
- "The diamagnetic version of the experiment was acquired with 64 scans over a 1024 x 256 data point matrix, using 58 ms and 31 ms as acquisition parameters in the direct and indirect dimension, respectively."
- "The paramagnetic tailored experiment was optimized for the identification of fast relaxing signals, acquiring 2048 scans over a 400 x 160 data point matrix, using 31 ms and 22 ms as acquisition in the direct and indirect dimension, respectively."
- "Recycle delay and C'/N INEPT transfer length were taken as short as 200 ms and 8 ms, respectively. The short recycle delay was used to enhance signal

[Figure]

> intensity of peaks with $^{13}C'$ $R_1 > 5$ s$^{-1}$. The C'/N INEPT transfer was shortened from 12.5 ms to 8 ms to incorporate the IPAP module and account for fast relaxing signals affected by paramagnetic clusters. The 8 ms transfer delay provides a slightly lower efficiency of the IPAP $^1J$ C'-C$\alpha$ decoupling and gives rise, in principle, to incomplete suppression of doublet components. However, as we are dealing with broad signals, the effect is hidden by paramagnetic broadening. The efficiency of C'-N coherence transfer vs R$_2$ $^{13}C'$ relaxation is reported in Appendix A, figure A3."

All these aspects are now reported in the manuscript, and critically commented, so they could be exploited by other scientists in the application of this approach to other paramagnetic systems.

*Further minor issues:*

*According to the Material and Methods section $^{15}N$ relaxation experiments (R$_1$, R$_2$, hetNOE) were performed for the diamagnetic part of the protein. The purpose and result of these experiments is not reported at any point in the manuscript.*

We thank the reviewer for pointing this discrepancy out., The purpose of the $^{15}N$ relaxation measurements was to determine the quaternary structure of our "as purified" mitoNEET construct, evaluating the relaxation information only of the diamagnetic part of the protein. We found that the protein is in a dimeric state, as showed for others mitoNEET constructs. Indeed, the molecular tumbling resulted to be $11.6 \pm 0.8$ ns that is compatible with a dimeric form for a 9.7 kDa protein. The purpose and the results of the $^{15}N$ relaxation experiments have been now reported in the revised version of the manuscript.

*Proton 1D spectra were recorded with a spectral width of 320 ppm, much wider than required for spectral range observed here. Were any specialized wide-band pulses employed that would be able to excite a ~ 200-kHz region?*

The pulse used is the standard 90° with no specialized wide-band features. We used a room temperature, high-power, probe at 600 MHz that has a 90° as short as 7 us. This was enough to excite the signals of the oxidized form. We also recorded spectra, using shorter excitation pulses, to seek for signals over a 300 ppm spectral window, on the oxidized and reduced form. However, no signals have been observed above 70 ppm and therefore these spectra are not shown.

[Figure]

*It is mentioned that the CON experiment was optimized for paramagnetic systems (section 4.2.3). Which modifications were applied? Simply shorter magnetization transfer periods? An INEPT delay of 8 ms is specified in the experimental section, which is shorter than 1/(2\*1JC'Ca). Is that sufficient to incorporate the IPAP module?*

With tried several delays and the 8 ms transfer described here was the best compromise between incorporate the IPAP module and account for fast relaxing signals affected by paramagnetic clusters. An 8 ms evolution of $^1J$ C'-Ca is only 1 ms shorter than the standard delay used for IPAP. The slight less efficiency of the IPAP decoupling gives rise, in principle to incomplete suppression of doublet components. However, as we are dealing with broad signals, the effect is hampered by $R_{2para}$ effects. To better discuss these effects, we have added in Appendix, an additional figure (Figure A3) to monitor the efficiency of C'/N coherence transfer vs $R_2$ relaxation, which could be useful for deciding for the optimal delay depending on the relaxation properties of the investigated system. We reported this modification also in the paragraph 3.2.2.

Anonymous Referee #2

*In this study, Camponeschi et al use NMR to characterize mitoNEET, a mitochondrial $Fe_2S_2$ protein. By using 1D NOE experiments, $^{13}C$ direct-detected experiments, and the optimization of NMR experiments for paramagnetic systems, the authors show significantly reduction of the "blind" sphere of the protein around the paramagnetic cluster, thus allowing the detection of residues possibly involved in the biological function of mitoNEET. The study has significant implications in the fields of paramagnetic NMR and FeS proteins. Some revisions are recommended.*

We thank the reviewer for the comments and questions which allow us to better explain our work. We'll try to answer all the questions raised by the reviewer. Specifically:

*1. I have some general questions about the mitoNEET protein I hope the authors can help answer.*

*a) If mitoNEET can repair Fe-S proteins by donating its own $Fe_2S_2$ cluster, how does it reacquire the $Fe_2S_2$ cluster? Can the authors comment on the source of its $Fe_2S_2$ cluster?*

The source of mitoNEET cluster is still unknown. Ferecatu and coworkers (see Ferecatu et al., JBC, 2014, 289, 41, 28070-28086) demonstrated that the origin of iron and sulfur moieties required for mitoNEET maturation is mitochondrial, and that

several components of the mitochondrial iron sulfur cluster (ISC) assembly and export machineries, such as ISCU, FXN, NFS1, HSC20, and ABCb7 are essential for the assembly of a [Fe$_2$S$_2$] cluster on mitoNEET, whereas early and late acting components of the cytosolic iron sulfur cluster assembly (CIA) machinery are not. However, the mechanism of mitoNEET cluster maturation was not clarified, and, to the best of our knowledge, a specific protein able to repair mitoNEET cluster has not been identified yet.

*b) The redox states of mitoNEET are crucial for its function and stability. How are the redox states of mitoNEET regulated in cells?*

Although *in vivo* studies addressing how the redox states of mitoNEET are regulated in the cell are still missing, several *in vitro* studies showed that mitoNEET clusters can be reduced by many cellular reductants. Indeed, mitoNEET redox state can be regulated *in vitro* by biological thiols such as reduced glutathione (GSH), L-cysteine, and *N*-acetyl-L-cysteine (Landry AP, Ding H, *J Biol Chem* 2014, 289, 4307–4315), human glutathione reductase (Landry AP et al *Free Radic Biol Med*. 2015, 81, 119–127), reduced flavin nucleotides (Landry AP et al. *Free Radic Biol Med* 2017, 102, 240–247; Tasnim H et al. *Free Radic Biol Med* 2020, 156, 11–19) and human anamorsin (Camponeschi F et al *JACS*, 2017, 139, 9479–9482), while NAD(P)H and NADH are not able to reduce mitoNEET clusters (Landry AP, Ding H *J Biol Chem* 2014, 289, 4307–4315). This aspect is now addressed in section 2 of the manuscript (lines 107-113).

*2. Some experimental details are needed.*

*a) For M9 media growth, how much ($^{15}$NH$_4$)$_2$SO$_4$ and $^{13}$C-glucose were supplemented?*

M9 media were supplemented with 1 g of ($^{15}$NH$_4$)$_2$SO$_4$ and 3 g $^{13}$C-glucose per liter. This is now added in the Materials and Method section.

*b) What kind of anaerobic environment was used?*

The protein was purified and handled inside an inert gas glove box, working with O$_2$ < 1 ppm. This is now specified in the Materials and Methods section.

*c) Does the phosphate buffer contain any NaCl?*

No, it doesn't

[Figure]

*d) I assume there were additional steps to remove the extra K₄Fe(CN)₆ or sodium dithionite?*

K$_4$Fe(CN)$_6$/sodium dithionite were removed after oxidation/reduction of the cluster using a PD10 desalting column. This detail has been added to the Material and Methods section. Thanks for the comment.

*2) What's the Fe₂S₂: protein ratio 'as purified'? It would be helpful to include UV data to show the load of Fe₂S₂ on the protein in both redox states.*

Non-heme iron and acid-labile sulfide quantification data (not reported in the manuscript) obtained for anaerobically purified mitoNEET following a previously published procedure (Banci L. et al. *Chem. Biol.* 2011, *18*, 794–804), gave 2.0 ± 0.1 Fe/mitoNEET and 1.9 ± 0.2 S$^{2-}$/mitoNEET (mol/mol of monomeric protein; error is the standard deviation of 4 measurements), meaning that we purified mitoNEET with ~ one [Fe$_2$S$_2$] cluster per monomer. A new figure, (Figure A1) has been added to the manuscript to show UV-visible data for both oxidation states.

*3) The authors purified the protein in an anaerobic environment, I assume it's because the Fe₂S₂ is susceptible to oxidative damage. Would addition of 10mM K₄Fe(CN)₆ to the protein solution damage the Fe₂S₂ cluster?*

In order to avoid oxidation of the [Fe$_2$S$_2$]$^+$ clusters of mitoNEET or oxidative damage of the [Fe$_2$S$_2$]$^{2+}$ clusters of mitoNEET upon exposure to O$_2$, we worked in an anaerobic environment using an inert gas glove box. This ensured long term stability of mitoNEET [Fe$_2$S$_2$]$^{+/2+}$ clusters. Indeed, during the NMR experiments we didn't observe changes in the cluster-bound protons signals in the paramagnetic NMR experiments or changes in the HN amide backbone signals in the diamagnetic and paramagnetic $^1$H-$^{15}$N experiments. Indeed, when the [Fe$_2$S$_2$] cluster of mitoNEET is disassembled the protein undergoes a folded-unfolded conformational change and the HSQC spectrum of the protein changes significantly (Ferecatu et al., JBC, 2014, 289, 41, 28070-28086; Golinelli-Cohen et al. *J Biol Chem.* 2016, 291, 7583–7593). Such changes were not observed in the HSQC spectra of reduced or oxidized mitoNEET over a period of roughly 12 h, suggesting that the cluster is stably bound to the protein for all the NMR experimental time.

The same behavior was observed when 10 mM K$_4$Fe(CN)$_6$ was added to the protein solution and the removed by PD10. Indeed, it can be stated that damaging of the [Fe$_2$S$_2$] cluster was not observed upon addition of K$_4$Fe(CN)$_6$.

*4) Is the purified mitoNEET protein a homodimer as shown in Fig. 1A?*

Yes, the protein was purified as a homodimer, as suggested by size exclusion chromatography data and by [15]N relaxation measurements. Indeed, the latter account for a $\tau_R$ value of $11.6 \pm 0.8$ ns, which is consistent with a dimeric state of the protein, whose molecular weight is ~18 kDa. [15]N relaxation data have been reported in appendix A, Figure A4, in the revised version of the manuscript.

*5) In Fig. 1A, can the authors highlight the residues that are affected by different redox states?*

As suggested by the reviewer, we have added a new figure in Appendix A (figure A2) were the residues affected by the different redox states are highlighted. These residues belong to the inter-subunit region as pointed out also by the reviewer. Residues number involved in the redox switch are 45, 48, 49, 53, 55, 56, 57, 58, 60, 63, 64, 65, 69, 94, 95, 96, 97, 103.

*6) Fig 1B, how were the chemical shift differences between two redox states calculated?*

The chemical shifts differences have been calculated using the following equation.

$\Delta_{HN} = ((\delta_H)^2 + (\delta_N/5)^2)^{1/2}$. This is now included in the Materials and Method Section in the revised version of the manuscript.

*7) It's intriguing to me that the redox state change would mainly affect the regions involved in inter-subunit contacts. Do the authors have any hypothesis why?*

We completely agree, it is very intriguing and interesting that the region affected by the redox state change is the inter-subunit one. Our hypothesis is that, in order to perform its function, mitoNEET has to switch between different conformational states, with the redox state change being one of the ways of regulating these transitions. Indeed, when mitoNEET passes from the "inactive", reduced state to the "active", oxidized state it might adopt a less tight conformation that facilitates the cluster transfer to IRP1 or to other apo recipient proteins, possibly driven by higher solvent accessibility of the cluster itself. We have added this comment to the discussion (lines 504-510).

*8) There is no mention of Fig. 1C in the text. The author might add some.*

[Figure]

Figure 1C (now figure 1B) is now mentioned in the manuscript according to the suggestion of the reviewer.

*9) Can the authors provide some explanations why no hyperfine shifted signals were observed for the reduced [Fe$_2$S$_2$]$^+$-bound form of mitoNEET?*

As already reported in literature, this a typical effect in mammalian [Fe$_2$S$_2$]$^+$, in particular in the case of the two irons ion pairs with delocalized valence. This has been first described by J Markley and coworkers and interpreted as due to the fundamentally different patterns of electron delocalization observed, for reduced [Fe$_2$S$_2$]$^+$ centers in plant and vertebrate ferredoxins (Skjeldal et al, Biochemistry. 1991; 30 (37), 9078-9083). When valence is delocalized, the iron ions have much slower electron spin relaxation rates than in the localized valence pairs, thus determining much broader lines often undetectable for $^1$H signals and eventually detectable, as very broad signals, only by $^2$H NMR measurements (Xia et al, Archives Biochem, Biophys, 2000, 373 (2), 328-334.) Actually, these aspects have been addressed in the discussion section in which we added a further comment on this point.

*10) The authors should provide the data showing the broadening of signal B collected in D$_2$O.*

A figure showing the broadening of signal B in D$_2$O has been added in appendix A (figure A5).

*11) The authors might want to highlight the additional residues assigned by $^{15}$N-IR-HSQC-AP in the structure of mitoNEET.*

Actually, IR-HSQC-AP and CON experiments pointed out a number of resonances, unobserved in the diamagnetic experiments that belong to the residues in the proximity of the cluster. However, the sequence specific assignment of these resonances, requires a quantitative analysis of R$_1$ and R$_2$ H$_N$ and H$_C$ rates, a series of triple resonance experiments optimized to provide scalar connectivities, $^{13}$C paramagnetic HSQC data and an "a-la-carte" analysis in order to identify the scalar and dipolar connectivities to confirm the assignment. This is beyond the aim of this work.

*12) The labels in Fig.3 are too small to read, the authors might want to improve that.*

[Figure]

We have modified the figure with larger labels. Thanks for the comment.

Anonymous Referee #3

*Camponeschi and her co-workers described a synergic application of paramagnetic and diamagnetic NMR techniques on protein mitoNEET, a dimer iron-sulfur protein, in both oxidation states. The NMR signals from residues surrounding the metal cofactor is usually crucial for understanding the structure-function in Fe-S proteins and is also challenging to detect due to the paramagnetic cluster. The authors demonstrate how to combine different paramagnetic NMR methods including 1D NOE, paramagnetism-tailored HSQC experiments, 13C detection experiments to reveal the information of protons as close as 4-5 Å around the paramagnetic cluster. The information obtained offers insights into the unique electronic properties of mitoNEET, which help to understand the role of the electronic structure in the biological function of NEET protein. The work in fact provides a potential general protocol that could be applied on many other similar challenging systems. The author gave a nice introduction on the history of NMR study of Fe-S protein started from 1970, and one that of paramagnetic NMR applications. The NMR data were elucidated and presented clearly; the manuscript is well written as well.*

*One concern is, what is new here for those paramagnetic techniques? The author may want to make it clearer in the paper. I recommend the paper to be published with changes to emphasize more on technical advances that applied here.*

We thank the reviewer for his/her comments. In the revised version we have essentially re-written the paragraph 3.2.2, we have added five Figures to the manuscript, and we have added new paragraphs in the abstract (lines 22-26), in Section 2 (lines 141-144), in the results Section (lines 354-361; 390-396) and in the Conclusions (lines 518-521) in order to include more details on the experimental aspects as well as on the specific advances that we have used for the characterization of this protein. As we have already discussed following comments of Anonymous referee #1 and yours, the novelty of our work relies on the analysis and interpretation of the paramagnetic NMR spectra of mitoNEET, never reported before and which are different to those reported for previously investigated $[Fe_2S_2]^{2+/+}$ proteins. The paramagnetic NMR spectra of mitoNEET in both oxidation states, provided a detailed description of its unique electronic properties, that is important for the understanding of the biological function

of the protein. Moreover, the characterization of mitoNEET gave us the opportunity to review the NMR spectra of $[Fe_2S_2]^{2+}$-containing proteins and to underline the subtle but significant differences, among them. We also thought that the special contribution to Rob Kaptein could have been an excellent opportunity to review on how the NMR characterization of FeS Proteins and NMR methodological implementations have been linked very closely and we thank the reviewer for appreciating our ideas.

As already pointed out in our reply to reviewer 1, we believe that the mitoNEET case is a very nice example of how a protocol based on the combination of various experimental approaches tailored to paramagnetic systems spanning from the more recent IR-HSQC-AP to the "ancient" 1D NOEs, could provide insights into the knowledge of a challenging system of high biological interest.

*Specific comments:*

*1.    Line 230, Figure 1(b), The chemical shift differences are all positive, it looks the chemical shift differences are absolute values of amide H only? The author may describe how to obtain these values. It might also be interesting to map these residues with significantly shift difference to the structure.*

Values reported in figure 1C indeed represent absolute values, obtained as $\Delta_{HN} = ((\delta_H)^2 + (\delta_N/5)^2)^{1/2}$. This is now included in the Materials and Method Section in the revised version of the manuscript.

*2.    Line 256: Figure 2, right, it might be better to label the cluster binding residues in figure.*

Done. Thank you for the comment.

*3.    346, "peak labelled with asterisk" are difficult to recognize in the figure, some are labeled "+" or "x".*

We modified both Figure 3 and caption to make them clearer.

*4.    Some minor format issues:*

*a.    Line 172, D$_2$O*

Done

[Figure]

*b.    Line 174, 177,…, name of experiments not consistent: eg. HNCACO or HN(CA)CO?*

Done

*c.    Line 210, $T_{1max}$*

Done

---

## Author Response (AR2)

*Florence, March 16th, 2021*

*Professor Gottfried Otting*
*Editor of*
*Magnetic Resonance*

TITLE: The long-standing relationship between Paramagnetic NMR and Iron-Sulfur proteins: the mitoNEET example. An old method for new stories or the other way around?
AUTHORS: Francesca Camponeschi, Angelo Gallo, Mario Piccioli and Lucia Banci

Dear Gottfried,

Herewith enclosed please find the revised version of the manuscript, where we have addressed all points raised by you and by reviewer #1, and that we have carefully checked for typos and inconsistencies.

Editor

*Lines 215 and 219: superfluous commas*

Done

*Line 360: 'refs Ciofi 2014' is not an acceptable format of citation.*

Done

Anonymous Referee #1

*In the current revised manuscript Camponeschi and co-workers have significantly enhanced the description of their approach to elucidate the electronic properties of mitoNEET. In particular, the previously almost unreadable NMR experimental paragraph 3.2.2. has been extensively re-written and the emphasis in the Results section has been laid on modifications required to study paramagnetic proteins by solution NMR. Some additional information and visualization of results is now provided in the Appendix.*
*In principle my original caveat concerning the novelty of each individual method used here still applies. However, as the paper, apart from its significant biological implications, now represents a nice "tutorial" for the investigation of challenging paramagnetic proteins the editors may consider it for publication in Magnetic Resonance.*

[Figure]

*Minor points:*

*p6, bottom: "Chemicals shifts data…" should be replaced by "Chemical shift data…."*

Done

*Fig. 1: I suppose the lower panel, in which parts b and c have been interchanged, replaces the upper panel. The legend should be changed accordingly.*

Done

*Fig. A3: Why was 20 Hz used for 1JC'N in the calculation of the transfer efficiency of the CON experiment, rather than the more realistic average value of 15 Hz?*

We have now used 15 Hz to calculate the transfer functions in Figure A3 in Appendix A.